



# Historical and future changes in global flood magnitude – evidence
# from a model-observation investigation
Hong Xuan Do[(1)(2)(3)(*)], Fang Zhao[(4)(5)(*)], Seth Westra[(1)], Michael Leonard[(1)], Lukas Gudmundsson[(6)],
Jinfeng Chang[(7)], Philippe Ciais[(7)], Dieter Gerten[(5)(8)], Simon N. Gosling[(9)], Hannes Müller Schmied[(10)(11)],
Tobias Stacke[(12)], Boulange Julien Eric Stanislas[(13)], Yoshihide Wada[(14)].
(1) School of Civil, Environmental and Mining Engineering, University of Adelaide, Adelaide, Australia.
(2) Faculty of Environment and Natural Resources, Nong Lam University, Ho Chi Minh City, Vietnam.
(3) School for Environment and Sustainability, University of Michigan, Ann Arbor, Michigan, United States.
(4) School of Geographical Sciences, East China Normal University, Shanghai, China.
(5) Potsdam Institute for Climate Impact Research, Potsdam, Germany.
(6) Institute for Atmospheric and Climate Science, Department of Environmental Systems Science, ETH Zurich, Zurich,
Switzerland.
(7) Laboratoire des Sciences du Climat et de l'Environnement, CEA-CNRS-UVSQ/IPSL, Université Paris Saclay, 91191 Gif sur
Yvette, France.
(8) Geography Dept., Humboldt-Universität zu Berlin, Berlin, Germany.
(9) School of Geography, University of Nottingham, Nottingham, United Kingdom.
(10) Institute of Physical Geography, Goethe University Frankfurt, Frankfurt am Main, Germany.
(11) Senckenberg Leibnitz Biodiversity and Climate Research Centre (SBiK-F), Frankfurt am Main, Germany.
(12) Max Planck Institute for Meteorology, Hamburg, Germany.
(13) Center for Global Environmental Research, Japan.
(14) International Institute for Applied Systems Analysis, Laxenburg, Austria.
*(*) Corresponding authors:* Hong Xuan Do (hong.do@adelaide.edu.au) *and* Fang Zhao (fangzhao@pik-potsdam.de)
**Abstract.** To improve the understanding of trends in extreme flows related to flood events at the global scale, historical and
future changes of annual maximum streamflow are investigated, using a comprehensive streamflow archive and six global
hydrological models. The models' capacity to characterise trends in annual maximum streamflow at the continental and global
scale is evaluated across 3,666 river gauge locations over the period from 1971 to 2005, focusing on four aspects of trends: (i)
mean, (ii) standard deviation, (iii) percentage of locations showing significant trends and (iv) spatial pattern. Compared to
observed trends, simulated trends driven by observed climate forcing generally have a higher mean, lower spread, and a similar
percentage of locations showing significant trends. Models show a moderate capacity to simulate spatial patterns of historical
trends, with approximately only 12-25% of the spatial variance of observed trends across all gauge stations accounted for by the
simulations. Interestingly, there are significant differences between trends simulated by GHMs forced with historical climate and
forced by bias corrected climate model output during the historical period, suggesting the important role of the stochastic natural
(decadal, inter-annual) climate variability. Significant differences were found in simulated flood trends when averaged only at
gauged locations compared to when averaged across all simulated grid cells, highlighting the potential for bias toward well-
observed regions in the state-of-understanding of changes in floods. Future climate projections (simulated under RCP2.6 and
RCP6.0 greenhouse gas concentration scenario) suggest a potentially high level of change in individual regions, with up to 35%
of cells showing a statistically significant trend (increase or decrease) and greater changes indicated for the higher concentration
pathway. Importantly, the observed streamflow database under-samples the percentage of high-risk locations under RCP6.0
greenhouse gas concentration scenario by more than an order of magnitude (0.9% compared to 11.7%). This finding indicates a
highly uncertain future for both flood-prone communities and decision makers in the context of climate change.





## 1 Introduction

Global hydrological models (GHMs) are critical tools for diagnosing factors of rising trends in flood risk (Munich Re, 2015;Swiss Re, 2015;Miao, 2018;Smith, 2003;Guha-Sapir et al., 2015;CRED, 2015), and can help identify the contribution of changing flood hazard characteristics relative to the changing exposure of human assets to floods. GHMs are also used to project future changes in flood hazard, owing to their ability to simulate streamflow under projected atmospheric forcing. Using GHM simulations, several studies have found more regions showing increasing trends than decreasing trends in flood hazards at the global scale, and have attributed these changes to anthropogenic climate change (Dankers et al., 2014;Arnell and Gosling, 2014;Alfieri et al., 2015;Kettner et al., 2018;Willner et al., 2018;Asadieh and Krakauer, 2017). The pattern of increasing trends obtained from GHM simulations is consistent with observations of increases in precipitation extremes (Westra et al., 2013;Westra et al., 2014;Donat et al., 2013;Guerreiro et al., 2018) that have been used by a number of studies as a proxy to suggest that flood hazard may increase as a result of climate change (Alfieri et al., 2017;Pall et al., 2011;IPCC, 2012;Forzieri et al., 2016).

The inference of changes in flood hazard following the same direction as extreme precipitation may be appropriate over specific regions (Hoegh-Guldberg et al., 2018;Mallakpour and Villarini, 2015;Mangini et al., 2018), but recent evidence based on instrumental trends in flood hazard suggests it is not necessarily globally applicable. This is due to a 'dichotomous relationship' between trends exhibited in extreme precipitation and extreme streamflow (Sharma et al., 2018), highlighted in recent observation-based studies of trends in streamflow magnitudes (Wasko and Sharma, 2017;Do et al., 2017;Hodgkins et al., 2017;Gudmundsson et al., 2019). The hypothesised reason for this potentially inconsistent relationship is the complexity of the drivers of flood risk (Johnson et al., 2016;Blöschl et al., 2017;Berghuijs et al., 2016), with the implication that historical and future changes to flood hazard at the global scale are unlikely to be reflected by changes to a single proxy variable alone, such as annual maximum rainfall. For example, even though trends in extreme flows are highly correlated to changes in extreme rainfall when rainfall plays the dominant role (Mallakpour and Villarini, 2015;Blöschl et al., 2017), snowmelt-related flood magnitude has been found to decrease in a warmer climate, potentially due to a shift in snowmelt timing (Burn and Whitfield, 2016;Cunderlik and Ouarda, 2009). The sign of change is also unclear for locations where antecedence soil moisture plays an important role (Woldemeskel and Sharma, 2016;Sharma et al., 2018), owing to the combined influences of seasonal/annual precipitation, potential evaporation and extreme precipitation (Bennett et al., 2018;Ivancic and Shaw, 2015;Leonard et al., 2008;Wasko and Nathan, 2019).

To better understand historical and future trends in streamflow, the emphasis has therefore moved to analysing trends directly in streamflow measurements. Investigations using streamflow observations at global, continental and regional scales (see Do et al. (2017) and references therein) have generally detected a mixed pattern of trends, with some global-scale studies finding more stations having decreasing trends than increasing trends (Do et al., 2017;Hodgkins et al., 2017;Kundzewicz et al., 2004). These





conclusions appear *prima facie* to be inconsistent with model-based evidence, which generally suggests the opposite (more
locations showing increasing trends). However, varying sampling strategies, statistical techniques and reference periods make it
difficult to derive a common perspective of trends in global flood hazards from a composite of observational and modelling
studies. In addition, data coverage limitations (Hannah et al., 2011;Gupta et al., 2014;Do et al., 2018b) remain a barrier to reliably
benchmarking trends over some areas such as the flood-prone regions of South and East Asia.
GHMs, with the advantage of better spatial coverage, remain an important line of evidence about historical and future trends.
GHMs also enable 'factorial' experiments to explore the individual roles of atmospheric forcing, land use change and other
drivers of change on streamflow trends. However, unlike climate models, for which the performance in terms of reproducing
trends of extreme precipitation has been evaluated substantially (Kiktev et al., 2003;Kiktev et al., 2007;Kumar et al.,
2013;Sakaguchi et al., 2012), the performance of GHMs has been assessed mostly on their capacity to represent physical features
of the hydrological regime, such as streamflow percentiles, the seasonal cycle or the timing of peak discharge (Gudmundsson et
al., 2012a;Zaherpour et al., 2018;Beck et al., 2017;Zhao et al., 2017;Veldkamp et al., 2018;Pokhrel et al., 2012;Biemans et al.,
2011;Giuntoli et al., 2018). Streamflow variability can be subject not only to long-term changes in atmospheric forcing, but also
to climate variability (e.g. inter-annual, inter-decadal) as well as human activities across the drainage basin (Zhang et al.,
2015;Zhan et al., 2012). Thus, the GHMs' capacity to represent physical features of a hydrological regime is not necessarily
sufficient to determine their performance in simulating characteristics of trends in extremes.
To better understand the capacity of GHMs in simulating historical trends in extreme streamflow and potential implications for
the development of projections, this study focusses on three research objectives. The first objective is to evaluate the capacity of
GHMs, available at http://www.isimip.org through the Inter-Sectoral Impact Model Intercomparison Project ISIMIP phase 2a and
2b (Warszawski et al., 2014), to simulate trends in observed streamflow extremes during the 1971-2005 historical period. The
particular interest is in reconciling observed and simulated trends in historical streamflow extremes at the global and continental
scale using the Global Streamflow Indices and Metadata (GSIM) archive (Do et al., 2018a;Gudmundsson et al., 2018b), to-date
the largest possible streamflow observations database. GSIM has been used in recent global scale investigations and is also an
important source for the production of GRUN, a data-driven century long runoff reconstruction (Ghiggi et al., 2019). The second
objective is to determine the representativeness of observation locations (streamflow gauges) in GHM simulations by comparing
trends simulated at these locations to trends simulated across all land grid points of GHMs. This objective is motivated by the
sparse coverage of streamflow observations over several regions (e.g. South and East Asia), which could lead to biased inferences
over large spatial domains wherever gauges are not a representative sample. The third and final objective is to assess the
implication of model uncertainty for projections of flood hazard, focusing on the uncertainty of the mean/spread of trends together
with the spatial pattern of trends in annual maximum streamflow.



## 2 Data and methods

### 2.1 Observed and simulated streamflow datasets

The GSIM archive is used as daily observational discharge for this analysis. Daily streamflow simulations available through the ISIMIP are used, with historical simulations (ISIMIP2a) spanning from 1971 to 2005 (Gosling et al., 2019) and future simulations (ISIMIP2b) covering 2006-2099 period (Frieler et al., 2017). Six GHMs are considered: H08 (Hanasaki et al., 2008a, b), LPJmL (Schaphoff et al., 2013), MPI-HM (Stacke and Hagemann, 2012), ORCHIDEE (Guimberteau et al., 2014;Guimberteau et al., 2018), PCR-GLOBWB (Wada et al., 2014;Sutanudjaja et al., 2018), and WaterGAP (Müller Schmied et al., 2014;Mueller Schmied et al., 2016). To assess the model structural uncertainty across GHMs, trends in streamflow extremes simulated under observational atmospheric forcing, available through the Global Soil Wetness Project Phase 3 (GSWP3) reanalysis (Kim, 2017), were compared to observed trends. The influence of the acknowledged high uncertainty in climate models (Kumar et al., 2013;Kiktev et al., 2003) on streamflow simulations was assessed by comparing observed trends and trends simulated when using atmospheric forcing from four General Circulation Models (GCMs) for the historical period ('hindcast' simulations). These GCM were bias corrected but their simulations have different sub-monthly, inter-annual and decadal variability and thus the hindcast simulations reflect both GHM and GCM uncertainty. To quantify the implication of model uncertainty for future projections of flood hazard, trends simulated under projected climate change by the end of this century (using the same four GCMs) were also assessed. As a result, four simulation settings were used in this study, denoted by the atmospheric forcing; an overview is given in Table 1. These settings comprise two historical runs (GSWP3 and GCMHIND runs), and two future runs (GCMRCP2.6 and GCMRCP6.0), collectively amounting to a total of 69 simulations (see Table S2 in supplementary with full list of simulations).

For GSWP3 simulations, naturalised runs (i.e. human water management not taken into account) were chosen, since this setting is available for more GHMs when compared to the human impact setting (i.e. human water management inputs were used). A preliminary analysis (see section 4 of supplementary material) shows that both 'naturalised runs' and 'human impact runs' exhibit similar characteristic of trends in peak discharge. Although significant efforts were made by ISIMIP to keep the setting across simulations as consistent as possible, there were some differences in model versions and input data (e.g. WaterGAP was used in ISIMIP2a while WaterGAP2 was used in ISIMIP2b; ORCHIDEE (Guimberteau et al., 2014) was used in ISIMIP2a while ORCHIDEE-MICT (Guimberteau et al., 2018), with improvements on high latitude processes, was used in ISIMIP2b). As a result, there are potential effects of technical discrepancies to the findings which cannot be checked in the context of this study. In addition, owing to technical requirements across GHMs, the number of land grid cells with available data is also different across simulations.



**Table 1.** Summary of streamflow observation and simulation datasets used in this study. GSIM was used as the observed streamflow database. Streamflow simulations were obtained from six GHMs (H08, LJPmL, MPI-HM, ORCHIDEE, PCR-GLOBWB and WaterGAP). One observational atmospheric forcing dataset (GSWP3) and outputs of four GCMs were used as input for streamflow simulations.

| Reference window | Streamflow obs./sim. | No. of GCM-GHM combination | Description | Note |
|---|---|---|---|---|
| Historical (1971-2005) | GSIM | - | Observational streamflow selected from GSIM archive. | Streamflow daily observations for 3,666 unique locations |
| | GSWP3 (ISIMIP 2a) | 6 | Historical simulation forced by observational atmospheric forcing. | Model did not use human water management input. |
| | GCMHIND (ISIMIP 2b) | 21 | Historical simulation using atmospheric forcing from four GCMs: GFDL-ESM2M, HadGEM2-ES, IPSL-CM5A-LR and MIROC5. | |
| Projection (2006-2099) | GCMRCP2.6 (ISIMIP 2b) | 21 | Future simulation forced by projected atmospheric forcing under greenhouse gas concentration scenario RCP2.6. Four GCMs were used: GFDL-ESM2M, HadGEM2-ES, IPSL-CM5A-LR and MIROC5. | No HadGEM2-ES simulation for MPI-HM. |
| | GCMRCP6.0 (ISIMIP 2b) | 21 | Future simulation forced by projected atmospheric forcing under greenhouse gas concentration scenario RCP6.0. Four GCMs were used: GFDL-ESM2M, HadGEM2-ES, IPSL-CM5A-LR and MIROC5. | No HadGEM2-ES and MIROC5 simulations for ORCHIDEE. |

## 2.2 Simulated streamflow extraction and catchment selection for observation-model comparison

To enable an observation-model comparison, simulated discharge needs to be extracted from gridded model output. Large-scale hydrological models, however, generally do not simulate discharge accurately over small-to-medium size catchments due to the coarse resolution of river network datasets in their routing schemes (Hunger and Döll, 2008). To address this limitation, previous GHMs evaluations usually selected large catchments (a threshold of 9,000 km² was adopted, approximating the size of a one-degree longitude/latitude grid cell) and routed discharge (units: m³/s) at the outlet of the catchment was used as simulated streamflow for a specific catchment (Zhao et al., 2017;Veldkamp et al., 2018;Zaherpour et al., 2018;Liu et al., 2017;Zaherpour et al., 2019). For evaluation studies that used relatively small catchments (e.g. area less than 9,000 km²), the un-routed runoff



simulation (units: mm/day) was extracted while observed discharge was converted to runoff using catchment area prior to
comparison (Gudmundsson et al., 2012b;Beck et al., 2017). To increase the sample size for the model-observation comparison
(the first objective), the present study used both daily (i) un-routed runoff for small catchments and (ii) routed discharge
simulations for large ones, and thus two extraction procedures were adopted. A summary of these extraction procedures is
provided below while detailed technical descriptions are provided in section 2 of supplementary material.
• For catchments with area from 0 to 9,000 km$^2$: un-routed runoff (mm/day) was extracted and then converted into
discharge (m$^3$/s) by multiplying averaged runoff with catchment area. Specifically, catchment boundaries were
superimposed on the GHM grid to obtain the weighted-area tables, which were then used to derive averaged runoff from
the un-routed runoff simulation. To avoid double counting runoff from the same grid points, runoff for catchments that
share similar weighted-area tables (i.e. similar simulated streamflow would be extracted – see supplementary section 2
for detail description) was averaged (using catchment areas as weights) and a single 'averaged time series' was used in
place of the runoff from the component catchments.
• For catchments with area greater than 9,000 km$^2$: the 'discharge output' approach (Zhao et al., 2017) was adopted to
extract routed discharge (m$^3$/s) from the GHM cell corresponding to the outlet of each catchment.
To ensure sufficient data is available for historical trend analysis, only GSIM stations with at least 30 years of data available
during the 1971-2005 period were considered (each year having at least 335 days of available records). These relatively strict
selection criteria also enable a comparison between this study and preceding observation-based investigations (Gudmundsson et
al., 2019;Hodgkins et al., 2017). As catchment boundary shapefiles (Do et al., 2018b) were used to extract simulated streamflow
for small catchments, stations were further filtered using two criteria: (i) availability of reported catchment area, and (ii)
catchment boundary was accompanied with a "high" or "medium" quality flag (i.e. the discrepancy between reported and
estimated catchment area is less than 10%).
A total of 4,595 stations satisfied the quality selection criteria, of which large catchments (i.e. area greater than 9,000 km$^2$) where
no suitable grid cell could be identified were further removed (11 catchments). For cases of two or more small catchments (i.e.
area less than or equal to 9,000km$^2$) having similar weighted-area tables, the 'averaged time series' (using catchment areas as
weights) was calculated. A total number of 1,542 time series fell in this category and were aggregated into 624 'averaged time
series'. Figure 1 shows the spatial distribution of the final dataset for model-observation comparison, containing data for 3,666
locations (3,042 non-averaged time series and 624 averaged time series). The majority of available catchments are located in
North America and Europe, with some regions over Asia, Oceania and South America are also covered.



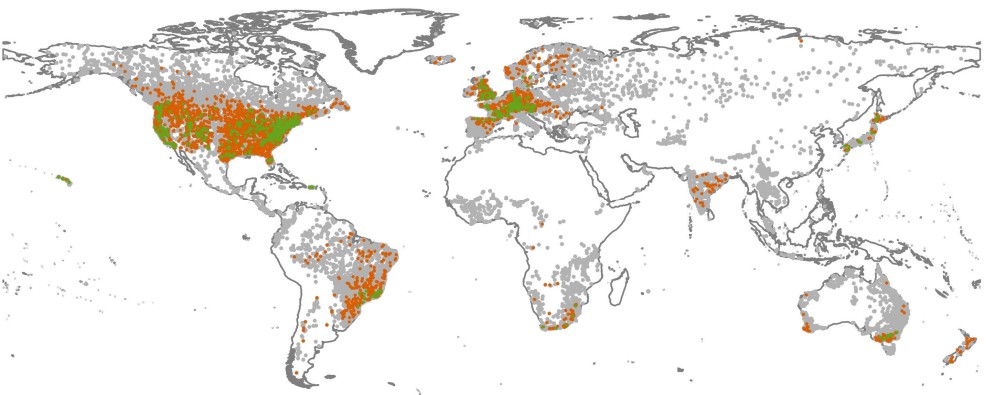


**Figure 1.** Locations of 3,666 streamflow observations (brown dots: 3,024 non-averaged time series; green dots: 624 averaged time series, where geographical coordinates were averaged from all component gauging coordinates) selected from GSIM archive for the model-observation comparison. Grey dots indicate GSIM time series that were removed due to insufficient data availability or quality.

## 2.3 Detecting trends in annual maximum streamflow

For each streamflow dataset, daily discharge was smoothed to 7-day averages to reduce variability in simulated streamflow, which can arise from the coarse routing parameters of GHMs (Dankers et al., 2014). The annual maximum time series of 7-day averaged discharge (labelled as the MAX7 index in the GSIM archive) was then derived to represent peak flow events. For gridded datasets, the 'centre averaged approach' (e.g. averaged streamflow of Jan 7th is the mean value of Jan 4 – 10th) was used (the common setting of the CDO software, freely available at https://code.mpimet.mpg.de/projects/cdo), and the MAX7 timeseries was therefore derived for each GSIM station using this same approach. As a result, the derived value of the MAX7 index is slightly different to the value available in the online version of GSIM (Gudmundsson et al., 2018a), which applied a 'backward-moving average' technique (e.g. averaged streamflow of Jan 7th is the mean value of Jan 1 – 7$^{th}$). Our preliminary analysis (not shown), however, indicated that this difference did not lead to substantial changes in the key findings.

The magnitude of trends in the MAX7 index at a specific catchment or grid cell was quantified using the normalised Theil-Sen slope (Gudmundsson et al., 2019;Stahl et al., 2010) and the results are expressed in % change per decade. The significance of the local trend was assessed using a Mann-Kendall test at the 10% two-sided significance level (Wilks, 2011). The null hypothesis (no trend) is rejected if the two-sided $p$-value of the test statistic (Kendall's $\tau$) is lower than 0.1, while the direction of the trend (i.e. increasing or decreasing) was determined using the sign of $\tau$.

## 2.4 Statistical techniques

To address the three identified objectives, trends in streamflow extremes obtained from GSIM (observed trends) and ISIMIP simulations (simulated trends) are analysed. The observed trends were available for 3,666 observation locations. Simulated trends



were available for all 59,033 GHM grid cells (estimated from routed discharge of each grid cell; Antarctica and Greenland were
removed). To enable a model-observation comparison, we also extract a subset of simulated trends over the 3,666 observation
locations (described in section 2.2).

### 2.4.1   A hypothesis-test approach for comparison of trend characteristics

A range of hypothesis tests (summarised in Table 2; GSWP3 simulations were used to assess GHM uncertainty while GCMHIND
simulations were used to assess the combined GCM-GHM uncertainty) was applied to address the first two objectives, which
require comparing trend characteristics exhibited from different streamflow datasets. Four characteristics of trends were assessed:
-   Trend mean: The mean (% change per decade) of trends in streamflow extremes across all gauge-/cell-based time series

over a spatial domain. A hypothesis test was adopted to assess whether the trend means exhibited from two specific

streamflow datasets (e.g. model vs. observed) are significantly different from each other.

-   Trend standard deviation: The standard deviation (% change per decade) of trends in streamflow extremes across all

gauge-/cell-based time series over a spatial domain. A hypothesis test was adopted to assess whether the trend of

standard deviations exhibited from two specific streamflow datasets are significantly different from each other.

-   Percentage of significant trends (%): The percentage of trends in a domain that are statistically significant, with gauge-

or cell-based significance calculated using the Mann-Kendall test at the 10% significance level. To assess whether the

percentage of significant (increasing/decreasing) trends exhibited from a specific streamflow dataset is produced by

random chance, a field significance test (Do et al., 2017) was adopted.

-   Trend spatial pattern: The spatial distribution of trends in streamflow extremes over a spatial domain. Pearson's (spatial)

correlation between trends of two datasets was used as a measure of similarity in the trend spatial structure. The

hypothesis test (pattern similarity test) was adopted to assess whether: (i) the correlation between simulated trends

introduced by GHMs and observed trends is significantly higher than zero; and (ii) the correlation between trends

simulated under hindcast atmospheric forcing and observed trends is significantly lower than that between trends

simulated under observational atmospheric forcing and observed trends.

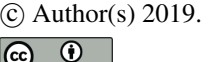



**Table 2.** Hypothesis tests conducted to address the first two objectives.

| Objective | Null-Hypotheses | Streamflow dataset | Statistical tests |
|---|---|---|---|
| Objective 1: Capacity of GHMs to reproduce observed trends in flood hazards | Hypothesis 1: Trend means obtained from two streamflow datasets over observation locations were not statistically different from each other. | (i) Observed discharge across 3,666 observation locations | Two-sample $t$-test at the 10% two-sided significance level |
| | Hypothesis 2: Trend standard deviations obtained from two streamflow datasets over observation locations were not statistically different from each other. | | Two-variance $F$-test at the 10% two-sided significance level |
| | Hypothesis 3: Percentage of significant trends obtained from all observation locations of a specific streamflow dataset was not produced by random chance. | (ii) Simulated discharge across 3,666 observation locations (extraction processes outlined in Section 2.2) | Field significance test similar to that presented in Do et al. (2017) was adopted. A moving-block-bootstrap (block-length $L = 2$) was used to derive a null-hypothesis distribution of the change that occurred due to random chance. The null hypothesis is rejected at 5% one-sided significance level when the true percentage falls on the right-hand side of the 95th percentile of the resampled distributions. |
| | Hypothesis 4: The correlation between trends obtained from two streamflow datasets was not significantly higher than '0' (i.e. zero pattern similarity). | | 'Zero pattern similarity' was compared to the probability distribution function (PDF) of pairwise correlation between simulated and observed trends, drawn from a bootstrap procedure similar to that proposed by Kiktev et al. (2003). The null hypothesis is rejected at 5% one-sided significance level when zero correlation falls on the left-hand side of the 5th percentile of the resampled distributions. |
| | Hypothesis 5: The correlation between | | The actual pairwise correlation between GCMHIND simulated trends |



Hydrology and Earth System Sciences Discussions Open Access

| | Hypothesis | Data | Statistical test |
|---|---|---|---|
| | | | and observed trends (denoted by $r_{GCMHIND}$) was compared to the bootstrapped PDF of correlation exhibited from GSWP3 simulated trends (denoted by $r^{*}_{GSW}$). If $r_{GCMHIND}$ falls on the left-hand side of the $5^{th}$ percentile $r^{*}_{GSWP3}$, there is evidence to reject the null-hypothesis at the 5% one-sided significance level. |
| **Objective 2:** The representativeness of observation locations in the GHM simulations | Hypothesis 6: Trend mean obtained from observation locations was not statistically different to that obtained from all grid cells. | (i) Simulated discharge across 3,666 observation locations (extraction processes outlined in Section 2.2) | Two-sample $t$-test at the 10% two-sided significance level |
| | Hypothesis 7: Trend standard deviation obtained from observation locations was not statistically different to that obtained from all grid cells. | | Two-variance $F$-test at the 10% two-sided significance level |
| | Hypothesis 8: Percentage of significant trends obtained from all grid cells of a specific streamflow dataset was not produced by random chance. | (ii) Routed discharge across all landmass grid cells (59,033 cells) | Field significance test similar to that presented in Hypothesis 3 but trends obtained from all grid cells were the subject of the assessment. |



### 2.4.2 Estimating uncertainty of trend characteristics across ensemble members


The third and final objective, which focused on the implications of GCM-GHM uncertainty on projected changes in
flood hazard, was addressed by quantifying the spread of trend characteristics (i.e. trend mean, trend standard
deviation, and percentage of significant trends) exhibited from routed discharge projections under two representative
concentration pathways.
The spatial uncertainty of projected trends (GCMRCP2.6 and GCMRCP6.0) was also quantified by calculating intra-
/inter-model correlation of the trend patterns across all ensemble members available under the two projections. Intra-
model correlation represents spatial uncertainty introduced by the GCM and was calculated from simulated trends
introduced by the same GHM (using different simulated atmospheric forcing). Inter-model correlation represents the
combined GCM-GHM spatial uncertainty, and was calculated for each pair of simulated trends that were: (i)
introduced by the different GHMs; and (ii) forced with different projected atmospheric forcing. This assessment also
identified regions that were consistently detected with a significant increasing trend across at least 11 simulations,
which can be used as an indication of potential 'hot-spots' of future flood hazard.
To assess the robustness of GHMs in projecting changes in flood hazard, each grid-cell of the discharge simulation
grid was then categorised into one of the five 'flood-risk' groups based on the number of GCMRCP2.6/GCMRCP6.0
simulation members projecting a significant increasing trend (Group 1: no members, Group 2: from 1 to 5 members,
Group 3: from 6 to 10 members, Group 4: from 11 to 15 members and Group 5: from 16 to 18 members). Each GSIM
gauge was also allocated into one of these five groups based on the gauge's geographical coordinates. The allocation
of gauges into these groups was then analysed to determine whether the most comprehensive global database of daily
streamflow records to-date was evenly distributed across the five 'flood risk regions'.

## 3 Results and Discussion


### 3.1 Capacity of GHMs to reproduce observed trends in flood hazards


Visual inspection of the normalised Theil-Sen slope across the GSIM time series (top panel of Figure 2; regional
maps provided in Supplementary Figure S4) shows a spatial pattern that is consistent with recent findings on trends in
observed flood magnitude (Mangini et al., 2018;Do et al., 2017;Mallakpour and Villarini, 2015;Gudmundsson et al.,
2019;Burn and Whitfield, 2018;Ishak et al., 2013). Specifically, decreasing trends tend to dominate Asia (most
stations located in Japan and India), Australia, the Mediterranean, western/north-eastern US and northern Brazil,
while increasing trends appear mostly over central North America, southern Brazil and northern Europe (including
the UK). Note that the observation locations are not evenly distributed (86% in North America and Europe), and thus
the confidence of this assessment varies substantially across continents.





The multi-model average of GSWP3 simulated trends (trends simulated under observational atmospheric forcing;
middle panel of Figure 2) has generally good capacity to reproduce spatial patterns of observed trends. The multi-
model average of GCMHIND simulated trends (trends simulated under hindcast atmospheric forcing; lower panel of
Figure 2), however, could not reproduce some spatial agglomerations of trends in streamflow maxima (e.g. the
decreasing trends in south-eastern Australia, increasing trends over north-eastern Europe). This feature indicates the
inconsistent climate variability between GCMs and the real world, suggesting GCM climate forcing cannot account
for observed trends at sub-continental scale. In addition, GCMs uncertainty can potentially contribute to this
inconsitency. Interestingly, the multi-model average of both GSWP3 and GCMHIND simulations generally exhibits a
lower magnitude of changes (i.e. closer to 'zero change') compared to the observed trends. This feature is more
prominent in GCMHIND (21 simulations available) compared to GSWP3 (six simulations available), and can be
explained by two possibilities. The first possible explanation is the nature of averaging, which tends to smooth out
variability in trend magnitude across ensemble members, leading to a relatively 'close to zero' change across the
globe (given that each GCMs has stochastic decadal climate variability, so that averaging GCMs tends to cancel
trends). An alternative explanation is that individual simulations also exhibit a lower magnitude of change relative to
observation, which is not visible through Figure 2.
To further explore GHMs' performance, a more detailed comparative analysis between observed trends and
individual simulated trends using both historical climate forcings (via GSWP3) and GCM hindcasts was conducted.
Specifically, four characteristics of trends in extreme flows (i.e. trend mean, trend standard deviation, percentage of
significant trends and trend spatial structure) were assessed for individual simulations and the results are reported in
following sections. At the global scale, GSIM observed trends exhibit a mean and standard deviation of -2.4% and
9.9% change per decade over the 1971-2005 historical period. Furthermore, there are 7.5% (12.1%) stations showing
significant increasing (decreasing) trends (detected by the Mann-Kendall test at the 10% significance level). These
numbers, however, are not statistically significant at the global scale.

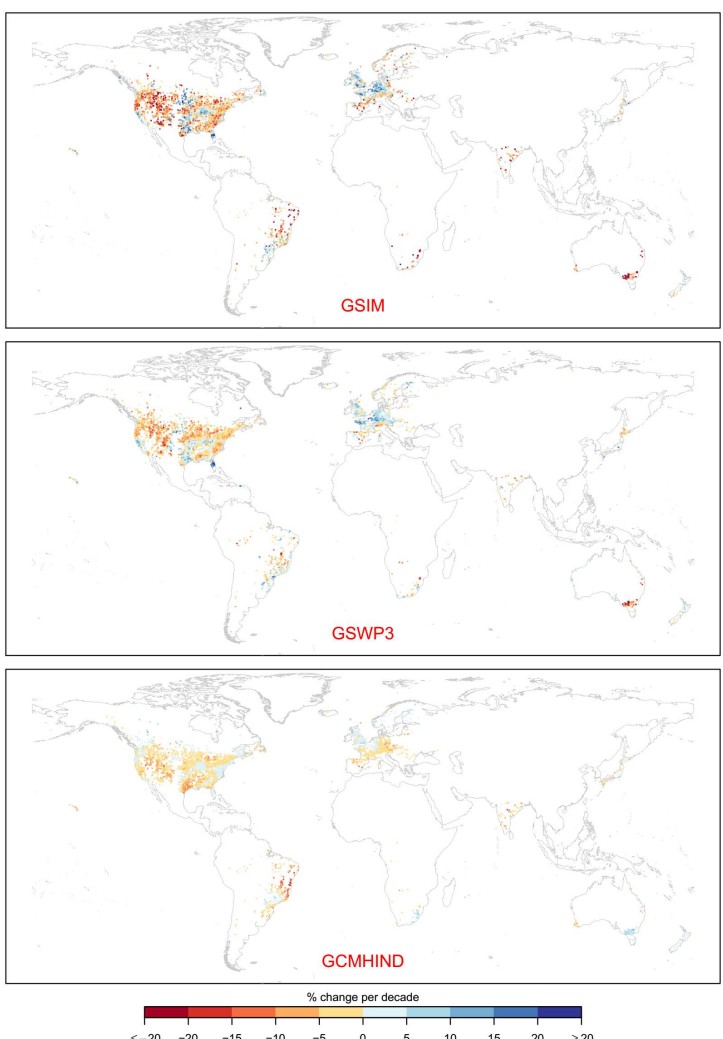


**Figure 2.** Normalised Theil-Sen slope for historical trends in flood magnitude (MAX7 index) exhibited over 3,666

locations across three streamflow datasets (top: GSIM; middle: GSWP3; bottom: GCMHIND). Multi-model average

is shown for simulated trends. Trend is expressed in % change per decade.


Table 3 shows the results of the global model-observation comparison using GSWP3 simulated trends across the six

GHMs. Compared to observed trends, most simulated trends have a significantly higher global trend mean at the

observed locations (ranging from -2.2% to 0.1% change per decade) and lower trend standard deviation (ranging from

7.1% to 8.7% change per decade). The percentage of locations showing significant trends varies substantially across

simulations, but the values were not statistically significant. All GHMs demonstrate moderate capacity in simulating

the spatial pattern of trends (spatial correlation coefficients range from 0.35 to 0.50, indicating that GSWP3 simulated





trends account for between 12%-25% of the cross-location variability in the observed trend signal). There is,
however, a notable difference in terms of the overall sign of trends simulated by each different GHM. This feature
indicates that using different GHMs can lead to different interpretations about the overall change in flood hazard at
the global scale, despite having a common boundary forcing. For example, PCR-GLOBWB suggests there are more
locations showing significant increasing trends (9.6%) than decreasing trends (6.1%) while LPJmL shows the
opposite pattern (4.5% and 7.3% of locations showing significant increasing and decreasing trends respectively). The
variation of trends characteristics exhibited by different GHMs also indicates that the 'closer to zero' trends of
ensemble averages (illustrated in Figure 2) likely reflects the implication of averaging rather than a systematic bias of
GHMs toward a low magnitude of change. As an implication, ensemble averages even though useful, should not be
used as a sole ground to infer change in floods, as this may undermine the actual magnitude of simulated trends.
**Table 3.** Characteristics of trends in the MAX7 index over the 1971-2005 period across 3,666 locations for GSIM
observed trends and GSWP3 simulated trends (six GHMs available). Trend mean and trend standard deviation are
expressed in % change per decade. Correlation was obtained from GSIM observed trends and GSWP3 simulated
trends for each GHM. Boldface texts represent values that reject the null-hypotheses outlined in Table 2 (hypothesis 1
to 4).

| GHM | Trend mean | Trend stand. dev. | % of sig. inc. trends | % of sig. dec. trends | Corr. obs. trend |
|---|---|---|---|---|---|
| H08 | **-1.9** | **8.3** | 4.8 | 6.7 | **0.42** |
| LPJmL | -2.2 | **7.1** | 4.5 | 7.3 | **0.37** |
| PCR-GLOBWB | **0.1** | **7.7** | 9.6 | 6.1 | **0.46** |
| WaterGAP2 | **-0.3** | **8.2** | 8.5 | 4.2 | **0.49** |
| MPI-HM | -2.1 | **8.7** | 5.6 | 7.5 | **0.50** |
| ORCHIDEE | **-1.4** | **8.6** | 7 | 8.2 | **0.35** |
| GSIM (observation) | -2.4 | 9.9 | 7.5 | 12.1 | - |


Table 4 provides the results of the model-observation comparison using GCMHIND simulated trends (intra-model
averages are shown while results of individual simulations are reported in section 4 of supplementary material).
Similar to GSWP3 trends, intra-model averages (i.e. calculated from simulations of one GHM) of GCMHIND trends
tend to have a higher global mean (ranging from -2.3% to -0.4% change per decade with 19 out of 21 simulations
suggesting a significantly different trend mean) and lower trend standard deviation (ranging from 7.4% to 8.7%
change per decade, with all simulations suggesting a significantly different trend standard deviation) than observed.
The composition between the percentages of locations showing significant trends varies substantially across
simulations (ranging from 2.2%/4.1% to 12.2%/17.3% for significant increasing/decreasing trends) and statistical
significance was found only for decreasing trends over three out of 21 simulations (two LPJmL simulations and one





MPI-HM simulation). The multi-model ranges encapsulate the observed trend mean and percentage of significant
trends, while the observed trend standard deviation is clearly above the range exhibited from all GCMHIND
simulations. The significantly lower simulated trend standard deviation can be partially attributable to the coarse
resolution of GHMs' atmospheric/land surface inputs, which may not sufficiently reflect the variation of hydrological
processes across small-to-medium size catchments.
**Table 4.** Characteristics of trends in the MAX7 index over the 1971-2005 period across 3,666 locations for
GCMHIND simulated trends. Trend mean and trend standard deviation are expressed in % change per decade. Intra-
model averages of trend characteristics are shown for each GHM. Values in the parentheses show the number of
simulations rejecting the null hypothesis (from 1 to 4) outlined in Table 2 (out of four GCMs). Multi-model
min/max/average values together with those exhibited from GSIM are also provided.

| GHM | Trend mean | Trend stand. dev. | % of sig. inc. trends | % of sig. dec. trends | Corr. obs. trend |
|---|---|---|---|---|---|
| H08 | -1.7 (4) | 8.5 (4) | 4.9 (0) | 8.8 (0) | 0.03 (2) |
| LPJmL | -2.3 (4) | 7.9 (4) | 4.2 (0) | 12.6 (2) | 0.09 (3) |
| PCR-GLOBWB | -1.1 (2) | 7.4 (4) | 7.5 (0) | 9.4 (0) | 0.06 (3) |
| WaterGAP2 | -1.3 (4) | 8.4 (4) | 5.4 (0) | 8.0 (0) | 0.02 (2) |
| MPI-HM | -1.8 (3) | 8.7 (3) | 5.7 (0) | 9.9 (1) | 0.05 (2) |
| ORCHIDEE | -0.4 (2) | 8.6 (2) | 6.9 (0) | 7.0 (0) | 0.04 (1) |
| Multi-model min | -4.2 | 7.0 | 2.2 | 4.1 | -0.06 |
| Multi-model max | 0.6 | 9.5 | 12.2 | 17.3 | 0.18 |
| Multi-model average | -1.5 | 8.2 | 5.6 | 9.5 | 0.05 |
| GSIM (observation) | -2.4 | 9.9 | 7.5 | 12.1 | - |


Among 21 GCMHIND simulations, the 'zero similarity' hypothesis (hypothesis 5) was rejected over 13 simulations,
indicating that GCM-GHM ensemble members possess some capacity to simulate the spatial structure of observed
trends in streamflow extremes. The correlation between GCMHIND simulated trends and GSIM observed trends
(ranging from -0.06 to 0.18), however, is significantly lower than that exhibited from GSWP3 simulated trends across
all GHMs (reported at Table 3). The results of the similarity assessment are illustrated for a single GHM (H08; as the
results were similar for other GHMs) in Figure 3, where the correlation between observed trends and GSWP3
simulated trends is significantly different from zero. In contrast, the correlation between observed trends and each of
the simulated trends under hindcast atmospheric forcing (GCMHIND simulations) is much lower, with two of the
four not being statistically higher than zero. These results confirm the substantial influence of atmospheric forcing on
the simulated trend pattern relative to GHMs structure.





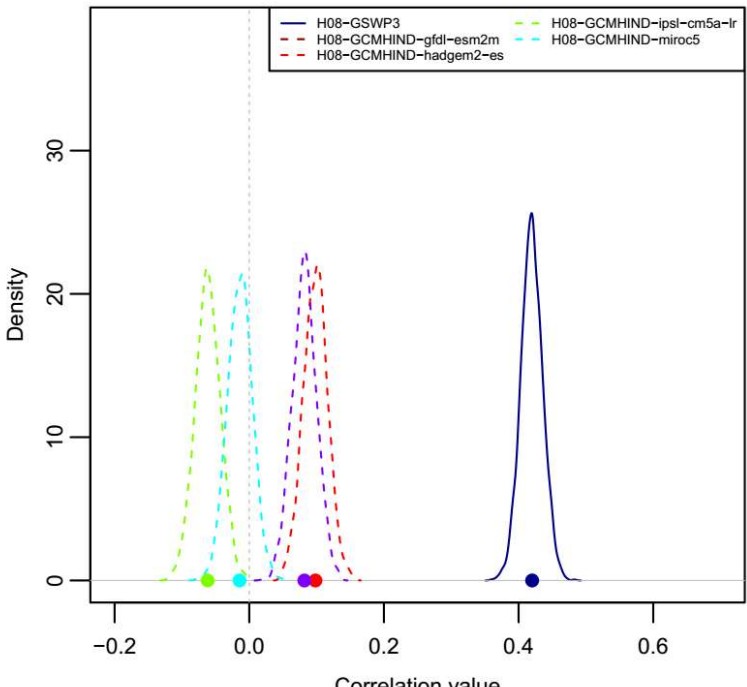

**Figure 3.** Model-observation correlation between observed trends and simulated trends across all simulations (GSWP3 and four GCMHIND simulations) of a single model (H08; similar results for other GHMs). Coloured dots indicate actual correlation between a specific simulated trend pattern and observed trend pattern across 3,666 locations. Colour lines represent the PDFs of correlation between simulated trend pattern and observed trend pattern obtained through a bootstrap resampling procedure ($B = 2000$).

To further quantify changes at the regional scale, a model-observation comparison (identical to that at the global scale) was conducted over six continents and the results are summarised in Table 5 (multi-model averages are shown). The trend mean exhibited from GSIM ranges from -10.7% (Oceania) to 2.4% change per decade (Europe) while trend standard deviation ranges from 8.3% (Europe) to 15.8% change per decade (Oceania). The percentage of significant increasing (decreasing) trends exhibited from GSIM ranges from 3.2% to 22.6% (from 6.3% to 29.1%) and the composition of significant trends across the six continents is consistent to a previous investigation (Do et al., 2017). The observed percentage of significant trends is found to be above random chance for Europe (increasing flood magnitude) and Australia (decreasing flood magnitude) and this feature is captured quite well by GSWP3 simulated trends, with at least half of the simulations confirming field significances detected from GSIM.





Similar to the assessment at the global scale, most GSWP3 simulations generally exhibit a higher trend mean
compared to the observed trend at the continental scale (see also Section 3.1 of the supplementary). Over data-
covered regions, a general lower trend standard deviation was also exhibited across all simulations, suggesting
substantial uncertainty of trends in streamflow extremes introduced by GHMs at the continental scale. The spatial
correlation is weakest in Asia, as no simulation rejects the null-hypothesis of 'zero similarity', while the spatial
correlation is strongest in Oceania (mainly southern Australia; correlation of 0.63). Oceania, however, exhibits the
highest model-observation discrepancy in trend mean and trend standard deviation, indicating the capacity of a given
GHM in terms of the trend spatial structure is not necessarily consistent with its performance in terms of the mean
and spread of trends.
GCMHIND simulations generally exhibit lower capacity in terms of reproducing trends. The majority of GCMHIND
simulated trends tends to not capture the continental trend mean and trend standard deviation exhibited in the
observed (see also Section 3.1 of the supplementary). GCMHIND trends also suggest the opposite composition
between percentages of significant trends compared to GSIM trends (e.g. simulated trends suggest more locations
showing significant increasing trends while observed trends suggest the opposite). Finally, the spatial correlation is
also significantly lower than GSWP3 correlation (except for Asia and South America). Among six continents,
GCMHIND trends exhibited the lowest correlation (-0.14) in Oceania, whereas GSWP3 suggested the strongest
correlation in this continent. This assessment further indicates the substantial impact of atmospheric forcing relative
to GHM model structure on the simulated trends in high flow events.





**Table 5.** Characteristics of trends exhibited from GSIM/GSWP3/GCMHIND streamflow dataset at the continental scale (each observation location of 3,666 sites was allocated into one of the six continents). For simulated trends, only the multi-model average is shown for each region. Trend mean and trend standard deviation are expressed in % change per decade. Values in the parentheses show the number of simulations rejecting the null-hypothesis described in Table 2 (up to six for GSWP3 simulations and 21 for GCMHIND simulations). For GSIM, field significance of increasing/decreasing trends was highlighted by boldface texts.

| Region | No. of loc. | Trend mean | | | Trend Stand. Dev. | | | % sig. inc. trends | | | % sig. dec. trends | | | Corr. obs. trends | |
|---|---|---|---|---|---|---|---|---|---|---|---|---|---|---|---|
| | | GSIM | GSWP3 | GCMHIND | GSIM | GSWP3 | GCMHIND | GSIM | GSWP3 | GCMHIND | GSIM | GSWP3 | GCMHIND | GSWP3 | GCMHIND |
| Asia | 96 | -3.1 | -1.2 (4) | -2.7 (6) | 8.8 | 6.6 (5) | 7.2 (15) | 4.2 | 4.2 (0) | 2.2 (0) | 15.6 | 10.3 (1) | 9.7 (2) | 0.07 (0) | 0.11 (11) |
| N. America | 2441 | -3.5 | -2.4 (3) | -1.6 (18) | 9.4 | 7.9 (6) | 8.0 (19) | 3.2 | 2.8 (0) | 5.3 (0) | 13.4 | 7.5 (0) | 9.3 (3) | 0.38 (6) | 0.03 (12) |
| Europe | 730 | 2.4 | 2.6 (6) | -0.7 (17) | 8.3 | 7.1 (5) | 5.9 (21) | **22.6** | 20.2 (3) | 7.3 (1) | 6.3 | 2.1 (0) | 10.1 (4) | 0.43 (6) | 0.10 (13) |
| Africa | 48 | -2.5 | -1.3 (0) | 1.5 (12) | 14.8 | 9.8 (5) | 8.0 (20) | 6.3 | 2.8 (0) | 9.6 (2) | 10.4 | 10.4 (0) | 3.3 (0) | 0.46 (6) | 0.07 (6) |
| S. America | 265 | -2.0 | -0.2 (5) | -3.6 (14) | 10.1 | 7.6 (6) | 10.0 (20) | 7.9 | 7.2 (0) | 3.4 (1) | 10.2 | 4.4 (0) | 13.4 (5) | 0.26 (6) | 0.18 (17) |
| Oceania | 86 | -10.7 | -6.1 (4) | 2.4 (21) | 15.8 | 10.9 (6) | 8.4 (21) | 4.7 | 3.7 (0) | 11 (2) | **29.1** | 22.1 (4) | 1.9 (0) | 0.63 (6) | -0.14 (2) |






### 3.2 Determining the representativeness of observation locations in the GHM simulations

To assess the representativeness of observations locations in GHM grid cells, trend characteristics obtained from all

simulated grid cells were compared to those estimated from the observation locations (3,666 sites globally). For

GSWP3 simulations, the results suggest a significant difference between trend characteristics from all model grid

cells compared to those obtained from the observation locations (Table 6; multi-model averages shown). This feature

is consistent at both global and continental scales, including North America and Europe – the continents with the best

stream-gauge density. Specifically, the trend mean tends to get closer to zero, while the trend standard deviation

obtained from all grid cells tends to be higher than that over observation locations. The difference between the

percentages of significant increasing/decreasing trends across all grid cells also gets smaller. For instance, the

percentage of observation locations showing significant increasing (decreasing) trends over Oceania is 3.7% (22.1%)

for GSWP3 multi-model averages (reported in Table 5), while the corresponding values are 10.7% (15.1%) when all

grid cells are considered (reported in Table 6). Additionally, field significance for increasing (decreasing) trends is

detected in two (four) out of six simulations over Oceania, while the same feature could not be detected over the

observation locations. These findings confirm that trends exhibited from observation locations are not a representative

sample of trends obtained from all simulation grid cells, which has also been suggested through Figure 1. As a result,

a common model-observation picture of changes in global flood hazard remains elusive. To enable a holistic

perspective of changes in extreme flows, it is therefore crucial to improve data accessibility and expand streamflow

observational networks to ensure unbiased samples are available for large scale investigations.

The findings using GCMHIND simulations are similar in terms of the trend mean (closer to zero) and trend standard

deviation (higher) across all grid cells relative to the observation locations. Across all land areas, the composition of

the percentages of land mass showing significant trends exhibited by GCMHIND simulations contradicts that

obtained from the GSWP3 simulations for many continents. For example, GSWP3 simulations suggest more land

areas showing significant decreasing trends than increasing trends over Asia and Oceania while GCMHIND

simulations indicate an overall increasing change in extreme flows over the same continents. This feature further

confirms the importance of atmospheric forcing in driving the spatial structure of the simulated trends, which will be

explored further in the next section.





**Table 6.** Characteristics of simulated trends across all grid cells at both continental and global scales (multi-model averages are showed). For each simulation, cell-based trend mean/trend standard deviation was compared to that of gauge-based trends (reported in Table 4). Values in parentheses represent the number of simulations reject the null-hypothesis described in Table 2 (up to six simulations for GSWP3 and 21 simulations for GCMHIND). GSIM results are also provided for reference.

| Region | Trend mean | | | Trend Stand. Dev. | | | % sig. inc. trends | | | % sig. dec. trends | | |
|---|---|---|---|---|---|---|---|---|---|---|---|---|
| | GSIM | GSWP3 | GCMHIND | GSIM | GSWP3 | GCMHIND | GSIM | GSWP3 | GCMHIND | GSIM | GSWP3 | GCMHIND |
| Asia | -3.1 | -0.7 (3) | 0.4 (16) | 8.8 | 10.3 (6) | 9.0 (15) | 4.2 | 7.7 (0) | 9.6 (7) | 15.6 | 9.9 (3) | 7.7 (4) |
| N. America | -3.5 | -1.8 (4) | 0.4 (19) | 9.4 | 10.3 (6) | 8.3 (17) | 3.2 | 6.9 (1) | 8.2 (4) | 13.4 | 12.3 (5) | 6.6 (0) |
| Europe | 2.4 | 1.1 (5) | 0.2 (16) | 8.3 | 8.5 (5) | 8.4 (20) | 22.6 | 11.5 (2) | 9.1 (5) | 6.3 | 4.5 (0) | 7.9 (3) |
| Africa | -2.5 | 0.7 (2) | -1.7 (15) | 14.8 | 11.0 (3) | 10.1 (12) | 6.3 | 10.9 (1) | 8.5 (6) | 10.4 | 11.2 (2) | 15.5 (11) |
| S. America | -2.0 | -2.0 (6) | -0.7 (19) | 10.1 | 8.7 (3) | 9.1 (17) | 7.9 | 4.9 (0) | 5.0 (0) | 10.2 | 8.6 (0) | 8.2 (1) |
| Oceania | -10.7 | -1.0 (6) | 0.5 (17) | 15.8 | 11.3 (4) | 10.4 (17) | 4.7 | 10.7 (0) | 10.3 (3) | 29.1 | 15.1 (1) | 9.6 (6) |
| Global | -2.4 | -0.6 (6) | -0.1 (20) | 9.9 | 10.3 (6) | 9.4 (19) | 7.5 | 8.3 (1) | 8.6 (6) | 12.1 | 10.2 (4) | 9.0 (6) |



### 3.3 The implication of simulation uncertainty on the projection of trends in flood hazard

This section focuses on the uncertainty in simulated trends under projected climate forcing at the global scale. For MPI-HM (no simulation for HadGEM2-ES forcing), streamflow was only simulated across the main stream-network (approximately 45% of the global land grid cells), and thus three simulations of this GHM were removed from the analysis. As a result, only 18 ensemble members were used to explore the uncertainty in projected trends (GCMRCP2.6 and GCMRCP6.0 – trends estimated for the 2006-2099 period and all cells were considered).

Table 7 shows a relatively low spread of the global trend mean (ranging from -1.3% to 0.8% change per decade; multi-model average of 0.0% change per decade for both GCMRCP2.6 and GCMRCP6.0) and trend standard deviation (ranging from 1.8% to 4.1% change per decade) across ensemble members. LPJmL and ORCHIDEE generally suggest a decreasing trend at the global scale, evident through the negative global mean and more grid cells showing significant decreasing trends. The standard deviation of trends in future simulations (multi-model average of 2.3% and 3.2% change per decade for GCMRCP2.6 and GCMRCP6.0 respectively) is substantially lower than the historical run (multi-model average of 9.4% change per decade as reported in Table 6). This feature is potentially due to the capacity of longer time series in capturing the inter-decadal variability of the streamflow regimes, with both dry and wet periods being considered (Hall et al., 2014). Projected trends under the RCP2.6 scenario generally have closer to zero mean and lower standard deviation compared to those introduced by the RCP6.0 scenario, reflecting the nature of an ambitious 'low-end warming' scenario, when anthropogenic climate change reaches its peak at the middle of the 21st century followed by a generally stable condition.

Interestingly, although most models suggest relatively moderate changes in the global trend mean, the composition between percentages of grid cells showing significant trends varies substantially, ranging from 7.5% (7.1%) to 30.1% (35.0%) for significant increasing (decreasing) trends at the 10 % level, with RCP6.0 generally exhibits higher values. This indicates that focusing on global averages may mask significant regional trends, as there was a substantially high percentage of locations exhibiting significant increasing and decreasing trends exhibited in individual models.

Uncertainty in the spatial structure of trends in streamflow extremes is further investigated using both intra-model (to reflect GCM uncertainty) and inter-model correlations (to reflect the combined GCM-GHM uncertainty). A more robust spatial pattern of projected trends under RCP6.0 was found, indicated through generally higher intra-/inter-model correlation (multi-model averages of 0.34/0.04) compared to those exhibited from trends simulated under RCP2.6; multi-model averages of 0.08/0.01) across all GHMs. This feature potentially reflects the less contrasted regional climate change of RCP2.6 relative to RCP6.0. The inter-model correlation (ranging from -0.18 to 0.21) is





consistently lower than intra-model correlation (ranging from –0.03 to 0.48) due to the combined uncertainty of both
GHMs and GCMs.
**Table 7.** The uncertainty in the characteristics of projected trends (GCMRCP2.6 and GCMRCP6.0) across 18
members at the global scale (five GHMs). Trend mean and trend standard deviation have unit of %-change per
decade. At-site significance of trend was identified using Mann-Kendall test at 10% level and the percentage of grid
cells showing significant increasing/decreasing trends was reported (no field significance test was conducted). Intra-
model average value of each metric across is shown for each GHM (numbers of simulations are provided in the first
column).

| Model | No. of sim | Trend mean | | Trend standard deviation | | % of sig. inc. trends | | % of sig. dec. trends | | Intra-model correlation | | Inter-model correlation | |
|---|---|---|---|---|---|---|---|---|---|---|---|---|---|
| | | *GCM RCP2.6* | *GCM RCP6.0* | *GCM RCP2.6* | *GCM RCP6.0* | *GCM RCP2.6* | *GCM RCP6.0* | *GCM RCP2.6* | *GCM RCP6.0* | *GCM RCP2.6* | *GCM RCP6.0* | *GCM RCP2.6* | *GCM RCP6.0* |
| H08 | 4 | 0.1 | 0.3 | 2.5 | 3.4 | 14.2 | 22.1 | 11.6 | 19.3 | 0.17 | 0.41 | 0.02 | 0.21 |
| LPJmL | 4 | -0.1 | -0.2 | 2.1 | 3.0 | 10.0 | 19.1 | 9.4 | 19.7 | 0.04 | 0.41 | 0.01 | 0.18 |
| ORCHIDEE | 2 | -0.5 | -0.8 | 2.6 | 3.6 | 9.1 | 14.4 | 17.6 | 28.1 | 0.07 | 0.34 | 0.03 | 0.11 |
| PCR-GLOBWB | 4 | 0.1 | 0.0 | 2.4 | 3.4 | 15.1 | 22.7 | 11.6 | 20.2 | 0.07 | 0.30 | 0.02 | 0.18 |
| WaterGAP2 | 4 | 0.2 | 0.5 | 2.3 | 3.0 | 13.0 | 25.9 | 8.0 | 11.8 | 0.03 | 0.25 | 0.01 | 0.17 |
| Multi-model min | - | -0.6 | -1.3 | 1.8 | 2.6 | 7.5 | 12.3 | 7.1 | 9.6 | -0.03 | 0.12 | -0.11 | -0.18 |
| Multi-model max | - | 0.4 | 0.8 | 2.9 | 4.1 | 18.0 | 30.1 | 21.2 | 35.0 | 0.30 | 0.48 | 0.21 | 0.21 |
| Multi-model average | - | 0.0 | 0.0 | 2.3 | 3.2 | 12.6 | 21.6 | 11.0 | 18.9 | 0.08 | 0.34 | 0.01 | 0.04 |


To quantity the robustness in terms of regions with significant trends in streamflow extremes, the number of
simulations showing significant increasing/decreasing trends was counted for each grid cell (value ranging from 0 to
18). As shown in Figure 4, the projections under RCP2.6 (top panels) do not suggest many regions with an increasing
trend for most ensemble members, but consistently suggest decreasing trends over the majority of Africa, Australia
and the western America. Although both scenarios suggested a similar spatial pattern, projections under the RCP6.0
scenario (lower panels) show a substantially higher robustness in terms of regions with significant changes over time
in streamflow extremes. For instance, significant increasing trends are projected consistently over southern and south-
eastern Asia, eastern Africa, and Siberia, while high agreement of decreasing trends is found over southern Australia,
north-eastern Europe, the Mediterranean and north-western North America. These findings share some similarity with
a previous investigation that used the ISIMIP Fast Track simulations (published before the ISIMIP2a and 2b
simulations used here) to identify regions projected with an increasing magnitude of 30-year return level of river flow
(Dankers et al., 2014). Specifically, both studies suggest overall: (1) an increasing trend over Siberia and South-East
Asia; and (2) a decreasing trend over north-eastern Europe and north-western North America. The present study,
however, additionally highlights a dominant decreasing trend over Australia, which was not shown previously. The
different numbers of ensemble members (45 in Dankers et al. (2014) and 18 in the present study) and greenhouse gas
concentration scenario (RCP8.5 in Dankers et al. (2014) and RCP2.6/RCP6.0 in the present study) between two


studies indicate that the choice of GCM-GHM ensemble and greenhouse gas concentration scenarios could lead to
substantially different projections of changes in flood hazard at the regional scale.

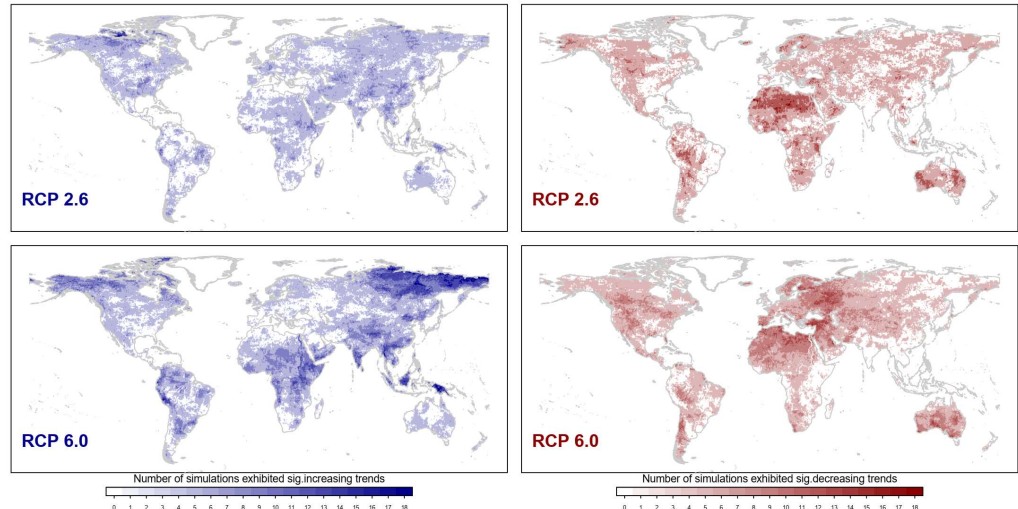

**Figure 4.** Number of simulations showing statistically significant trends at the 10% level at each grid cell. The left

panels show results for the assessment of increasing trends, while the right panels show results for significant

decreasing trends. Top: results of GCMRCP2.6 simulations; Bottom: results of GCMRCP6.0 simulations.

These results suggest the key role of GCM uncertainty in projections of changes in flood hazards, emphasising the
importance of a flexible adaptation strategy at the regional scale that can take this uncertainty into account (Dankers
et al., 2014). Such a strategy is achievable only through a reliable and robust understanding of the change in flood
hazard. The assessment of the representativeness of streamflow observations (section 3.2), however, demonstrated
that the observation locations selected for this assessment are not a representative sample of the entire land mass. As a
result, inference of changes in flood hazard may be biased toward well-observed regions.
To further highlight the potential impact of limitations in observed streamflow datasets, the proportion of available
stream gauges located in regions with different levels of projected 'flood risk' was assessed. We first categorised each
grid-cell into one of the five 'flood-risk' groups based on the number of simulations projecting a significant
increasing trend. In this analysis, RCP6.0 scenario was chosen as it yielded a higher global 'risk' of flood hazard
relative to RCP2.6 scenario. Figure 5 presents the percentage of all simulated grid cells (left panel) and of the subset
of GSIM station (right panel) falling in each of the five groups. As can be seen, 11.7% of grid cells fell into the "high
risk" groups (8.9% from Group 4 with 11-15 ensemble members, and 1.8% in Group 5 with 16-18 ensemble




members), compared to only 0.9% of stations available in GSIM archive (0.9% from Group 4 and no station located
in Group 5). In contrast, 68.9% of grid cells fell into the "low risk" groups (22.0% for Group 1 with no ensemble
members, and 46.9% for Group 2 with 1-5 ensemble members), compared to 89.5% of stations available in GSIM
archive (35.4% for Group 1 and 54.1% for Group 2). The uneven distribution of stream gauges indicates potential
difficulties in using observational records to provide an assessment of global or regional changes in flood hazard,
which in part arises from data caveats associated with the spatiotemporal coverage and quality of observed gauge
records across the globe.

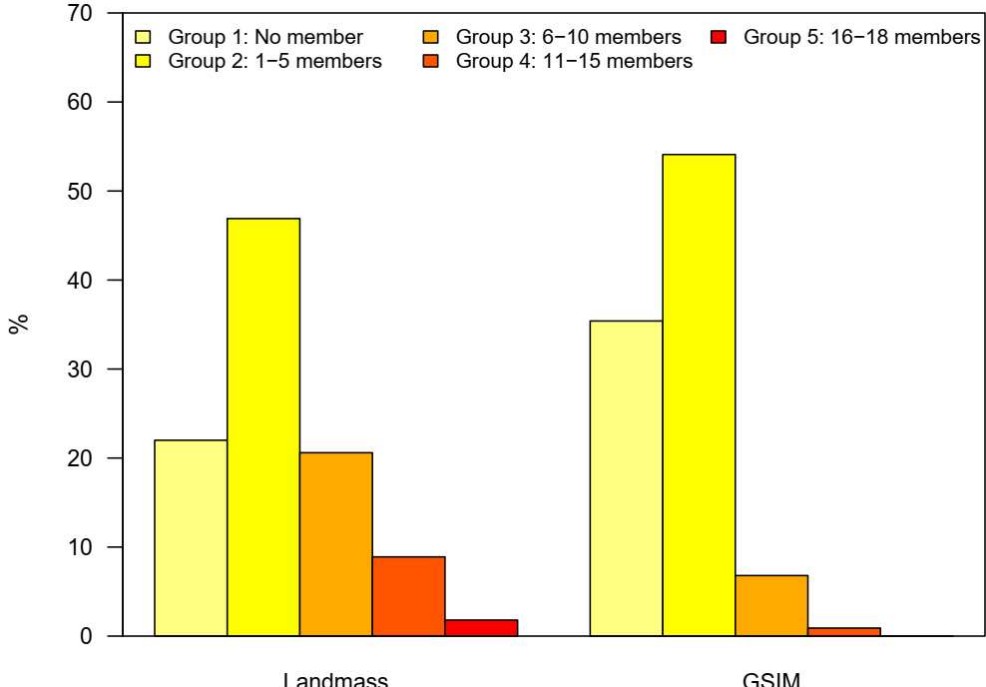


**Figure 5.** Percentage of grid-cell ("Landmass") grouped by the number of simulations projecting a significant
increasing trend under RCP6.0 scenario; and the percentage of streamflow stations ("GSIM") assigned into each
group. The range of possible simulations is from 0 to 18 and binned into five groups (Group 1: no members, Group 2:
from 1 to 5 members, Group 3: from 6 to 10 members, Group 4: from 11 to 15 members and Group 5: from 16 to 18
members). To identify which group a specific station belongs to, the geographical coordinates of that station was
superimposed on top of the global 'flood-risk' map.
**4 Summary and conclusions**
To reconcile observed and simulated trends in historical flood hazards at the global and continental scale, this study
evaluated the capacity of six GHMs to reproduce the characteristics of historical trends over the 1971-2005 period,



using observations from the Global Streamflow Indices and Metadata (GSIM) archive. The observed trends in annual
maximum streamflow confirm previous findings about changes in flood hazard over data-covered regions (Do et al.,
2017), in which significant decreasing trends were found mostly in Australia, the Mediterranean region, western US,
eastern Brazil and Asia (Japan and southern India), while significant increasing trends were more common over
central US, southern Brazil, and northern Europe.
The ability of GHMs to reproduce trends in streamflow maxima was assessed, focusing on four characteristics of
trends (i.e. the mean and standard deviation of trends, the percentage of stations showing significant
increasing/decreasing trends, and the spatial structure of trends). Trends simulated by GHMs, when using an
observational climate forcing, show moderate capacity to reproduce the characteristics of observed trends. Climate
forcing uncertainty (i.e. the effect of using different GCMs to simulate the historical climate), however, significantly
reduced the extent to which the GHMs' captured the observed spatial structure of trends. This was evident through
significantly lower spatial correlation between observed hydrological trends and simulated trends, when GCMs were
used for the climate forcing, than when climate observations were used.
The simulated trends over observed areas inadequately represented spatially averaged trends simulated for wider
spatial areas from all GHM grid cells at the continental and global scales. This was evident in most simulations for
trend mean and trend standard deviation, indicating a potential mismatch between observation-based and model-based
inferences about changes in flood hazard. As a result, alternatives for conventional approach in estimating change of
streamflow extremes at the global and regional scale (i.e. unweighted mean across all grid points) should be
investigated. For instance, the spatial weighted averages (e.g. using inverse distance relative to observed locations as
weights) could be used to compute global means of changes. Regional analysis using homogenised regions as the
basis of reporting spatial domains (Zaherpour et al., 2018;Gudmundsson et al., 2019) could be a potential alternative
for continental scale assessment.
Uncertainties of trends in streamflow extremes were analysed to assess their implication on the development of
projected changes in flood hazard over the 2006-2099 period. Under both RCP2.6 and RCP6.0 greenhouse gas
concentration scenarios, simulated trends across ensemble members have relatively low uncertainty in terms of the
global trend mean (ranging from -1.3% to 0.8% change per decade) and trend standard deviation (ranging from 1.8%
to 4.1% change per decade). The spread of the percentage of land mass showing significant trends is high, ranging
from 7.5% (7.1%) to 30.1% (35.0%) for significant increasing (decreasing) trends. This indicates that limited changes
to the global mean flood hazard could potentially mask out significant regional changes. The spatial correlations



across inter-model trend patterns are generally low (ranging from -0.18 to 0.21), further indicating high levels of
uncertainty.
In terms of regional planning to mitigate flood hazard, individual models may provide contradictory signals of
changes in flood hazard for a specific region. Under RCP6.0 scenario, some regions, e.g. south-eastern Asia, eastern
Africa, Siberia, were consistently projected with significant increasing trends, which has some similarity to previous
findings that used ISIMIP Fast Track simulations (Dankers et al., 2014). These 'high-risk' regions, however, are
sparsely sampled, covered by less than 1% of all available stream-gauges listed in the catalogue of GSIM. Data
coverage, as a result, remains the key limitation of this study, which could potentially lead to an erroneous conclusion
on the state-of-understanding of historical trends in flood hazard globally. Specifically, substantial changes, although
having occurred, might not be captured by available streamflow records.
Improved performance of GHMs in terms of simulating changes in flood hazard, considering the many factors
influencing model capacity, is achievable only through the combined efforts of many communities. The spread of
trends in streamflow extremes (trend standard deviation) could be simulated more accurately by finer spatiotemporal
resolution GHMs. Such an improvement in GHMs, however, is highly dependent on the quality of input datasets (e.g.
dam operations, historical irrigation databases and land-use/land-cover, in addition to atmospheric forcing), which are
driven by advances in other geophysical disciplines (Bierkens et al., 2015;Wood et al., 2011). The moderate capacity
of GHMs in terms of simulating the spatial structure of trends in streamflow extremes indicates the need for improved
representation of runoff generation at the global scale (e.g. to better reflect rainfall-runoff relationship and the
contribution of snow-dynamics), which is also a focus of large-sample hydrology (Gupta et al., 2014;Addor et al.,
2017). Uncertainty in GCMs, a long-standing challenge for the climate community, should also be addressed to
enable robust projections of flood hazard in a warmer climate. One possibility is through constraining model
performance using historical observations, which could potentially reduce the uncertainties of atmospheric forcing
projections (Greve et al., 2018;Lorenz et al., 2018;He and Soden, 2016;Padrón et al., 2019).
This study presents a comprehensive investigation of historical and future changes in flood hazard using a hybrid
model-observation approach. The results highlighted a substantial difference between trend characteristics simulated
by GHMs and that obtained from GSIM archive, suggesting more attention should be paid to investigating GHMs
performance in the context of historical and future flood hazard. This is particularly important to determine the
appropriateness of GHMs in specific investigations, as model performance may vary substantially across different
variables (e.g. moderate capacity in simulating spatial structure of trends may be accompanied by a low performance
in terms of simulating trend mean). Large-sample evaluations, however, are highly dependent on data availability,





which has been emphasised as one of the key barriers to a holistic perspective of changes in floods. Specifically, the
unevenly distributed GSIM stations, partially due to the constraint in data accessibility, do not provide representative
samples at both global and continental scale. Sustained and collective efforts from the broad hydrology community,
therefore, are required to make streamflow data becomes more FAIR (Findable, Accessible, Interoperable and
Reusable; see Wilkinson et al., 2016), and ultimately complement our limited understanding of flood hazards. Data
providers, considering their tremendous investments in maintaining and making streamflow observations available in
the public domain, remain key agencies to enhance the evidence-base of the global terrestrial water cycle and changes
in flood hazard. Centralised organisations such as GRDC or WMO should also push forward the movement of
making streamflow data accessible to the research community. More initiatives based on citizen science (Paul et al.,
2018) should be adopted, as this is a potential option to crowdsource water data and offset the limitation of traditional
observation system. Finally, attention should also be paid to stream gauges maintenance, data housekeeping and data
sharing to ensure ongoing flood monitoring is available to the present and future generations.
**Acknowledgement**
Hong Xuan Do is currently funded by School for Environment and Sustainability, University of Michigan through
grant number U064474. This work was supported with supercomputing resources provided by the Phoenix HPC
service at the University of Adelaide and Flux HPC service at the University of Michigan.

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
