# Peer review of "Historical and future changes in global flood magnitude – evidence"

_Hydrology and Earth System Sciences, 2019_

## Referee Comment (RC1) · Anonymous Referee #1 · 23 Aug 2019

Dear authors,

First of all, I want to congratulate you for this extensive piece of work. It clearly shows that a lot of work was carried out to analyse this huge load of simulated and observed streamflow data.

While I think that the content fits within the scope of HESS and the work makes use of recent literature, models, and data, the manuscript needs improvement before publication is possible.

Please see below my general and subsequently my specific comments.

General comments

- Not content related: the authors list reads like a "manel". Sad to see so little inclusion of other genders than male. Maybe something to think about for future studies?
- I do miss the overarching research motivation of this study. Even though the introduction contains a good amount of background literature and describes the three research objectives set, it does not become clear what the research gap is that the present study intends to fill. Or in short, how are the objectives derived and what is the (societal) relevance of the study? Please ensure this is clearly and concisely explained in the revised version of the manuscript.
- The models used in the study differ in quite some characteristics/schematization. An overview of these differences would be very useful (also possible as supplement). Besides, none of the deviations between results obtained with different GHMs is related to those different characteristics/schematization. I would guess that they do play a big role in explaining the obtained results and thus I recommend extending the manuscript with such an analysis (or a clear and convincing statement why not).
- The manuscript reads bit lengthy sometimes. To some extent, this is the result of reporting a lot of numbers, which are partially also provided within tables. My recommendation would be to let the tables and figures speak for themselves and to check which reported numbers can be neglected as they are not directly needed for understanding the methodology or results. Plus a check whether more concise wording could be used.
- This study is very much focussed on data and their analysis. Extending the implication of the findings of the study to the societal dimension would greatly benefit the manuscript to make the results more tangible and applicable.
- I would very much welcome it if at least the data pairs for the observation stations are provided via a supplement. This would be in the spirit of FAIR hydrologic modelling, thus increasing reproducibility of your findings.

Specific comments

- P1/L29+L30: An agreement of 12-25 % can hardly be named "moderate agreement", can it?
- P1/L31 "significant differences": please specify what kind of differences you refer to.
- P2/L52 "specific regions": specific regions such as? Please specify.
- P2/L53 "recent evidence": what evidence? Please provide name, source, etc.
- P2/L52-L66: what about the role humans play in changing flood hazard? Please add this dimension to the paragraph.
- P3/L77 "factorial experiments": what are you referring to with this term? Please explain or use more common terminology.
- P3/L87-L100: The description of the research objectives could profit from using bullet points
- P4/L105-L108: why are these models used? Why are some others available within ISIMIP not used? Please clarify.
- P4/L122-L126: As mentioned in the general comments, a (technical) description of the models is needed. A particular focus should be on how the models changed between ISIMIP2a and ISIMIP2b and whether those changes may have influence results (or not). Possible changes in e.g. functionality, spatial resolution, etc. may have had a great impact on results and thus affecting the comparison performed in your study. I thus strongly disagree that checking this is outside the context of the study.
- P6/L144-L147: what was the reason to not only use the un-routed runoff for all catchments? Wouldn't this increase comparability between results as it removes (unnecessary) transformation of results and units?
- P6/L149 "catchment area": which area estimates did you use? For all models the same? Per model based on catchment delineation? Please clarify to avoid that data was used inconsistently.
- P7/L177-L185: great you are pointing out the differences in methodology!
- P11/L228-L230: these lines read as if they should not be part of the methodology, rather of the results/discussion section. The fact that that there may be 'hot-spots' of future flood hazard should be discussed in more detail and thus deserves a more prominent location in the manuscript.
- P11/L231 "each grid-cell": each grid-cell or only those paired with a GSIM-location? Does not become very clear from reading.
- P11/L235-L237: why was this done? What does it add?
- P11/L240-L247: You describe the observation and simulations, but you do not mention the reasons behind it. Why are certain areas experiencing increases and others decreases? Is it all hydrology or not? Can we say something about the driving factors behind it? Please add.
- P12/L255+L256: What is the implication of this finding?
- P12/L261+L262: So, if it is not visible through Figure 2, how can it be an alternative explanation? This sounds contradictory to me – either it's possible based on your results or your results say it's not a thing. Please clarify.
- Figure 2: A bit bigger figure (maybe with subplots of USA, EU) would help seeing the differences between differences between historical trends.
- P14/L285-L287 and Table 3: what are possible reasons for the different model results? Model structure, processes simulated per model, spatial resolution, routing schemes applied or something else? Would be great if you could elaborate a bit on this.
- P14/L290+L291: if it should not be used as "sole ground", what other measures would you (like to) use to infer changes in floods?

- P23/L463+L464: Does that mean your results are not usable to help inform flood management practices in less well-observed areas? What would be the implications? Please elaborate briefly on the consequences of your results.
- P24/L475-L478: What would be ways forward to reduce the dependency on not evenly spatially distributed observation systems? Which opportunities do, for instance, remotely sensed data products bring? Please put your findings into context, here and/or the conclusions section.
- P26/L533-L537: It should be added that also the routing schemes of GHMs should improve, not only the runoff. Well timed runoff with right magnitude can still result if inaccurate streamflow if the routing scheme is too simplistic. Vice versa, higher-order routing schemes cannot perform at their best if input runoff is not accurate. Relevant literature: Hoch et al., 2019 (https://doi.org/10.5194/nhess-19-1723-2019) and Zhao et al., 2017 (https://doi.org/10.1088/1748-9326/aa7250).
- P27/L550-L559: I very much agree with this, well written!

---

## Referee Comment (RC2) · Anonymous Referee #2 · 24 Sep 2019

This manuscript describes the work of an extensive model study and in its final version will be for sure appreciated by the readers of HESS.

General Comments:

As the study presented is quite extensive, it is sometimes difficult to follow study setup and all the analysis steps. Therefore, the authors should provide a detailed schematic, showing the main building blocks of their study and the different steps of analysis (preferably showing the section numbers in the schematic as well) to allow the reader to have a complete 'picture' of the study design, before embarking on the details in the main text.

[Figure]

Additionally, due to the complexity of the study and details provided in the result section, I think a summary table or bullet points at the end of the study would be helpful for the reader to get a better overview of the key results obtained.

Another important point is that the study uses 7-day annual maximum as a surrogate for 'food'. This fact needs to be made more explicit throughout the study to avoid misunderstandings from the general perception of flood, which would shorter (e.g. often 1-day). This is of importance, as the results might be quite different. I.e. a single day peak value trend study will show different results, not only in terms of magnitude of change, but also in terms of the flood hydrograph shape. E.g. if floods would become flashier in some location in future, it might look as if the trend of a 7-day maximum might not change at all or get smaller, but the peak day could be of much higher magnitude. The authors need to make sure they call the variable under investigation for what it is, i.e. not calling it 'flood', 'peak discharge' or 'streamflow maximum' to avoid misunderstanding of the results. Along this line, I also think that the title '... changes in global flood magnitude ... ' is also misleading. The study shows rather an 'global assessment of the 7-day annual maximum average value'. Please consider changing the title to better represent the content of the study. Additionally, to avoid misinterpretations of your results please avoid using the term 'hazard' in its current form in the manuscript, as hazard means: hazard=risk*exposure (which is not the correct terminology here). The same also applies to the term 'risk' which is related to 'probability and consequences'.

In this manuscript I feel that the GHM are used by the authors as 'black-box' that give some output. However, for this study to be valuable, it would be important that the authors would try to relate the observed differences/deviations in the outputs to the actual differences in the hydrological model setup. The authors just state "... there are potential effects of technical discrepancies to the findings which cannot be checked in the context of this study" (L 126). However, I think based on the model selection, the authors should have a notion of why they selected certain models and what the

key differences are. Hence, the authors should at least try to come up (also based on past literature) with some sort of reasoning for model selection and also more importantly an interpretation of their findings... For example, are the changes the models are giving as an output considered in line with the current understanding of the effects of climate change on floods or are there surprising results? I think this could be done in a separate paragraph discussing/comparing with previous literature.

In several instances in the manuscript, the authors are highlighting the 'substantial influence of the atmospheric forcing in driving the spatial structure of the simulated trend'. I think this is another important point that needs to be discussed in more detail in the discussion section, i.e. why to the hydrological models have little influence....

Overall, I think a new separate discussion section of the results of such a complex analysis would be beneficial, as this would free up the room for a better refined summary and conclusion section, that focused on the key results and the overall implications of the results not just for the scientific world but also for the 'end-users', such as decision makers etc.

Specific Comments:

L37: For clarity, please provide significance level used in this study in parentheses.

L38: replace the term 'high-risk location'.

L54: Please provide reverence to this statement

L77: What is 'factorial evidence' in this regard? Please elaborate.

L121-122: Please elaborate why the authors think that the 'naturalised runs and the human impact runs exhibit similar characteristics of trend' Would one not expect considerable differences?

L126: What are the 'potential effects'. Can you briefly elaborate.

L127: Please also elaborate what the effects/impacts of this on the results are.

L158: What is the rationale of 335 days. Please explain briefly.

L172: Fig1: These colours are not 'safe' for colour-blind readers. Please use different colour combination

L184: 'Our preliminary analysis... did not lead to substantial changes'. So what were the 'not so substantial changes' one is wondering?

L192: Can you please name the 'three identified objectives' again as it is quite difficult to keep up with this extensive work.

L210: To spare the reader from having to go to the original reference, please name the field significance test used and elaborate briefly what exactly is evaluated.

L211: What 'Pearson's (spatial) correlation' was used? Reference? What variables are correlated?

L220: Please replace the term 'flood hazard' with something more appropriate to what has been done. This also applies to the subsequent usage, as well as the term 'flood-risk' later used in the manuscript.

L245 & 493:to me it does not look like norther Europe has increasing trends. Scandi-navia etc looks decreasing. ... Please check.

L258: I agree, very much with this point. The study analyses 'extremes (i.e. floods) but then model 'averages' are provided. His is counter intuitive. This can lead to strong underestimation of the actual changes. The usage of averages vs individual models that show extremes should be better discussed in the discussion section. Hence, I also agree with L 419.

L281: is this really 'the spatial pattern of trends' that is evaluated or is it a cell by cell comparison? Please elaborate and have in mind that although a correlation is it can still mean that the overall spatial pattern (i.e. approximate location of increasing and decreasing trends) might still be correct.

L 370-384: The authors mention 'a significant difference between trend characteristics from all model grid cells compared to those obtained from the observation locations' and the conclude that '' that trends exhibited from observation locations are not a representative sample of trends obtained from all simulation grid cells'' (L379-380) And then call ''to improve data accessibility and expand streamflow observational networks''. However, if there are such ''significant difference even in data rich regions, how can one justify expanding the network based on the previous finding? Instead to me this reasoning would rather require the need to improve our models instead (notwithstanding the fact that I agree with the data needs mentioned by the authors.)

L 460: Maye the authors can elaborate a little more what an 'flexible adaptation strategy' entails in terms of flood mitigation. Any suggestion on how this can be achieved under tight budgets. Can we as scientists not provide any guidance than just saying 'stay flexible' to those who have to take decisions know?

L531 & 534: Along the lines of improved GHM: It is not only important that the spatial patterns are being reproduced correctly but also that the timing of the high-flows/floods are being modeled correctly. I.e. 'the flood seasonality patterns can be used as an additional metric to test large-scale hydro-logical models for their ability to reproduce the spatial and temporal flood characteristics.' (Hall and Bloschl, 2018, HESS). As this would give more confidence that the models actually get the flood generation processes correctly.

L 538: What does 'constraining ' entail? Please briefly elaborate. Would this prevent the model to adjust to changes in the flood generating processes, as one would expect to happen in some regions of the world. E.g. from snow-melt floods to rainfall-generated floods?

L 550-559: I agree with this call, as this is very important. However, one needs to keep in mind that in many countries maintaining monitoring networks and data curation is/is considered too expensive. Hence it needs to be made clear to decision makers that

such data is of importance. However, I know of cases where countries/agencies have been or are currently considering discontinuing their data networks, as they don't see the benefit or don't see their data being used (partly lack of proper citation of the (often freely available) original data source). This implication needs to be kept in mind when large datasets of observational data are being compiled and subsequently only credit is given to the compiled data... This hides to the funding/responsible agencies the usage of their data (i.e. the original data source) and might lead to the misconception that their data is not being needed/downloaded and hence the data network can be discontinued and to allocate funds to more (perceived) useful sectors...

Fig S5: Suggest using same y-axis scale for all panels on the left/right to be able to compare the regions better with each another.

---

## Author Comment (AC2) · 30 Oct 2019

**Replies to reviewers – Submission titled "Historical and future changes in global flood magnitude – evidence from a model-observation investigation" [HESS-2019-388]**

**Replies to Reviewer #2**

We would like to thank Reviewer #2 for the constructive comments that will help us to improve the quality of the manuscript. For clarity, we formatted reviewer's comments in *blue, italic text*, while our responses are formatted in normal text.

*This manuscript describes the work of an extensive model study and in its final version will be for sure appreciated by the readers of HESS.*

We thank the reviewer for the encouraging evaluation.

*General Comments: As the study presented is quite extensive, it is sometimes difficult to follow study setup and all the analysis steps. Therefore, the authors should provide a detailed schematic, showing the main building blocks of their study and the different steps of analysis (preferably showing the section numbers in the schematic as well) to allow the reader to have a complete 'picture' of the study design, before embarking on the details in the main text.*

We will include an additional figure at the start of Section 2 to provide a complete picture of the analyses presented in this study.

*Additionally, due to the complexity of the study and details provided in the result section, I think a summary table or bullet points at the end of the study would be helpful for the reader to get a better overview of the key results obtained.*

Thank you for pointing out this issue from the reader's perspective. In our revision, we will carefully revisit the manuscript to improve readability and simplify the contents where relevant. We will also consider your suggestions to format the main findings in bullet points to communicate the key points better.

*Another important point is that the study uses 7-day annual maximum as a surrogate for 'food'. This fact needs to be made more explicit throughout the study to avoid misunderstandings from the general perception of flood, which would shorter (e.g. often 1-day). This is of importance, as the results might be quite different. I.e. a single day peak value trend study will show different results, not only in terms of magnitude of change, but also in terms of the flood hydrograph shape. E.g. if floods would become flashier in some location in future, it might look as if the trend of a 7-day maximum might not change at all or get smaller, but the peak day could be of much higher magnitude. The authors need to make sure they call the variable under investigation for what it is, i.e. not calling it 'flood', 'peak discharge' or 'streamflow maximum' to avoid misunderstanding of the results.*

We thank the reviewer for the suggestion. We would like to note that in our preliminary analysis, we also analysed the annual maximum values of daily streamflow (i.e. 1-day flood time series, MAX index). Although trend at specific site may vary between these two indices, we found that the key conclusions (e.g. the regional pattern of increasing/decreasing trends, the consistency between trends exhibited from observed data and that obtained from simulated data) are quite similar, regardless which index

being used (i.e. MAX or MAX7 index). To address your concerns, we will revise our manuscript substantially to ensure: (i) there is sufficient information about the consistency between results introduced by MAX and MAX7 index, and (ii) the terminologies are used appropriately (e.g. replace "streamflow maximum" by "7-day streamflow maximum").

*Along this line, I also think that the title '. . . changes in global flood magnitude . . . ' is also misleading. The study shows rather an 'global assessment of the 7-day annual maximum average value'. Please consider changing the title to better represent the content of the study.*

We acknowledge the reviewer's concern and will consider a change in the title. A possible option is to replace "magnitude" by "indicator"(i.e. "Historical and future changes in an indicator of global flood hazard - evidence from a model-observation investigation"), due to the fact that MAX7 can also be used as an indicator for floods, and using MAX index has generally led to comparable results to that of MAX7 index (discussed above). From our perspective, this proposed title is not misleading, and can potentially reach a broader readership than using a too technical title such as "Global changes in 7-day annual maximum average value".

*Additionally, to avoid misinterpretations of your results please avoid using the term 'hazard' in its current form in the manuscript, as hazard means: hazard=risk*exposure (which is not the correct terminology here). The same also applies to the term 'risk' which is related to 'probability and consequences'.*

We noted that this study was developed from the perspective that 'hazard' (e.g. flood magnitude, frequency or inundation) is a component of 'risk' (i.e. Flood Risk = Hazard x Values x Vulnerability; Kron, 2005). From this point of view, we judge 'hazard' the appropriate terminology to refer to the MAX7 streamflow index. We acknowledge the reviewer's concern about the use of the term 'risk' and will carefully evaluate the manuscript and clarify/make changes where relevant to ensure the appropriateness of each terminology.

*In this manuscript I feel that the GHM are used by the authors as 'black-box' that give some output. However, for this study to be valuable, it would be important that the authors would try to relate the observed differences/deviations in the outputs to the actual differences in the hydrological model setup. The authors just state ". . . there are potential effects of technical discrepancies to the findings which cannot be checked in the context of this study" (L 126).*

We agree with the reviewer that the relationship between the model's structure and model's capacity in simulating trends in floods is an important aspect. However, addressing this comment is not straightforward, as there are a total of six models with many factors (e.g. routing schemes, spatial resolution, and parameterisations) that could individually or collectively lead to output discrepancies. These aspects (i.e. possible reasons for different trends simulated by different GHMs) in fact is still under-represented in the literature. From our perspective, this type of investigation deserves a separate paper by itself as the work involved should be tremendous, and potentially involved another set of simulations (e.g. to check the sensitivity of simulated trends corresponding to changes in a specific factor).

In the revision, we will refine the introduction to clarify the key objectives of our study, which is to compare trends observed by different models and the uncertainty in projected trends rather than to explore the mechanisms driving discrepancies in model outputs. We will also highlight the reviewer's comment in the conclusion as a potential research direction.

*However, I think based on the model selection, the authors should have a notion of why they selected certain models and what the key differences are. Hence, the authors should at least try to come up (also based on past literature) with some sort of reasoning for model selection and also more importantly an interpretation of their findings. . .*

In this study, we did not make any model selections. Specifically, we used all hydrological models that have produced discharge outputs for both Phase 2a and 2b at the time this study was initiated (June 2018). In the revision, we will highlight this fact better to avoid confusion.

*For example, are the changes the models are giving as an output considered in line with the current understanding of the effects of climate change on floods or are there surprising results? I think this could be done in a separate paragraph discussing/comparing with previous literature.*

Our manuscript has highlighted that the historical trends obtained in the present study are consistent with what has been reported in the literature (Lines 240-247). The reviewer suggested that simulated trends should also be linked to the current understanding of the effects of climate change. Although this aspect is important, we intend to not cover it as our objectives are not to attribute change in flood hazard to climate change or human activities. For historical trends (1971-2005; or ISIMIP2a), the focus was to compare model capacity in reproducing observed trends and compare the performance of simulations driven by observed (GSWP3 simulations) and modelled atmospheric forcing (GCMHIND simulations). The ultimate goal is to show the uncertainty of trends in the MAX7 index detected from the current GHM-GCM ensemble. For future trends (2006-2099), the focus was on the robustness of projected trends introduced by the ensemble members.

In addition, there is another ISIMIP investigation dedicating on river flow changes attribution, and thus we decided to exclude this aspect from this manuscript to avoid overlap.

*In several instances in the manuscript, the authors are highlighting the 'substantial influence of the atmospheric forcing in driving the spatial structure of the simulated trend'. I think this is another important point that needs to be discussed in more detail in the discussion section, i.e. why to the hydrological models have little influence...*

The hindcast simulations of the global climate model are forced by historical $CO_2$ (Katragkou et al., 2015), and so the timing of wet/dry periods or the spatial distribution of precipitation will be different from what has been observed in the past. As precipitation is arguably one of the most important inputs for streamflow simulation, it is expectable that GCMHIND trends will have a more prominent impact on the spatial patterns of simulated trends relative to model structure. We will consider including this justification in the revision.

*Overall, I think a new separate discussion section of the results of such a complex analysis would be beneficial, as this would free up the room for a better refined summary and conclusion section, that focused on the key results and the overall implications of the results not just for the scientific world but also for the 'end-users', such as decision makers etc.*

Thank you for suggesting this potential improvement. We will revisit the whole paper to better discuss the findings and improve the paper readability. Some opportunities for improvement have been identified, which we believe will help the paper streamlined better:

- Revisit our introduction to clearly state the research objectives and narrate the analyses.

- Include an additional figure (in line with our previous response) to show the overall framework of the study and how does it address the research questions.

- Simplify the contents where relevant, potentially in Section 3.1, to exclude redundant information and make the analyses more focus.

We will also consider your suggestion (i.e. having a separate discussion section) during our revision.

**Specific Comments:**

*L37: For clarity, please provide significance level used in this study in parentheses.*

We will add the level of significance (10% two-sided) in the revision for clarity.

*L38: replace the term 'high-risk location'.*

Thanks for noting, we will evaluate the terminologies across the manuscript, potentially using "locations robustly projected with increasing flood hazards".

*L54: Please provide reverence to this statement*

The following sentences (Lines 54-62) in fact has extended our discussion and provided some evidence and references for this statement. We noted that this may be unclear and will revisit this paragraph to ensure the statement is justified.

*L77: What is 'factorial evidence' in this regard? Please elaborate.*

"Factorial experiments" refer to studies analysing the effect of different factors (e.g. land use change) on the response variable (e.g. changes in floods), as well as the effects of the interactions among the factors on the response variable. In the context of hydrological modelling, the impact of atmospheric forcing, land use change and human water management on streamflow trends could be "turn on/off" to provide a full "factorial experiment design". In the revision, we will revise this statement to improve clarity.

*L121-122: Please elaborate why the authors think that the 'naturalised runs and the human impact runs exhibit similar characteristics of trend' Would one not expect considerable differences?*

We thank the reviewer for the suggestion, which will be incorporated in our revision. Some potential reasons are the spatial distribution of stream gauges, which may be biased toward regions with insignificant changes in human intervention within the reference period (1971-2005), or the inclusions of small catchments (more that 3000 catchments with area less than 9000km$^2$), and floods are more sensitive to changes in extreme precipitation relative to the accumulated basin-wide influence of human impacts.

*L126: What are the 'potential effects'. Can you briefly elaborate.*

The most pronounced effect comes from the difference in the versions of GHMs that were used in ISIMIP2a and ISIMIP2b. Specifically, ISIMIP2a was designed as an evaluation framework to improve the models for the projection phase isimip2b. As a result, the assessment using historical simulation (from 1971-2005) may not reflect the "true" model capacity in simulating trends in floods during the future period (2006-2099). We will elaborate on this fact and the potential effects of different model versions

in the revision. However, as mentioned in our response to Reviewer#1, a solid conclusion about these effects may not be available.

*L127: Please also elaborate what the effects/impacts of this on the results are.*

Thanks for your comment. We will revise the manuscript to elaborate on the potential effects/impacts of technical differences across GHMs, potentially including:

- Different drainage direction maps across different models could lead to gauging stations (in some rare cases) that do not lie on the river network (Masaki et al., 2017).

- Different models do not have the same set of coastal cells which may lead to some minor effect to the statistics when averaged across all simulation grid-cells.

- ORCHIDEE runs on 1-degree resolution but is routed at 0.5-degree resolution and thus influenced by a stronger spatial averaging that could lead to more flatten discharge time series.

*L158: What is the rationale of 335 days. Please explain briefly.*

The rationale of this choice is every single year must have at least 90% of streamflow data available. This criterion is a common data filtering condition in large-scale observation-based investigation (Do et al., 2017;Mallakpour and Villarini, 2015). This data criterion was chosen to fit the purpose of a hybrid observation-simulation study. We will consider clarify this methodological choice in the revision.

*L172: Fig1: These colours are not 'safe' for colour-blind readers. Please use different colour combination*

We will revise this figure in the revision to address this concern. Specifically, we will consider the use of an eight-color discrete palette that is colorblind safe (available in ggthemes R package at https://rdrr.io/cran/ggthemes/man/colorblind.html).

*L184: 'Our preliminary analysis. . . did not lead to substantial changes'. So what were the 'not so substantial changes' one is wondering?*

The preliminary assessment showed that the regional patterns of changes detected from MAX and MAX7 indices are generally consistent. We will clarify this point in our revision.

*L192: Can you please name the 'three identified objectives' again as it is quite difficult to keep up with this extensive work.*

We will consider provide the identified objectives as bullet points to remind readers about the focus of this study.

*L210: To spare the reader from having to go to the original reference, please name the field significance test used and elaborate briefly what exactly is evaluated.*

We will incorporate your suggestions in the revision by adding a brief explanation about the bootstrapping technique that was used.

*L211: What 'Pearson's (spatial) correlation' was used? Reference? What variables are correlated?*

Here we computed the Pearson's correlation *r* metric (Kiktev et al., 2003;Galton, 1886) to represent the spatial consistency between two sets of trends in MAX7 index. We will clarify this statistical technique to improve clarity in the revision.

*L220: Please replace the term 'flood hazard' with something more appropriate to what has been done. This also applies to the subsequent usage, as well as the term 'floodrisk' later used in the manuscript.*

As mentioned in our previous response, we think flood hazard is the appropriate term to refer to the magnitude of MAX7 index. Nevertheless, we will carefully evaluate the manuscript to ensure the most appropriate terminologies are used.

*L245 & 493:to me it does not look like norther Europe has increasing trends. Scandinavia etc looks decreasing. . .. Please check.*

We thank the reviewer for noting out this mistake – which should be "the northern part of Western Europe". We will revise the manuscript to ensure correct description is presented.

*L258: I agree, very much with this point. The study analyses 'extremes (i.e. floods) but then model 'averages' are provided. His is counter intuitive. This can lead to strong underestimation of the actual changes. The usage of averages vs individual models that show extremes should be better discussed in the discussion section. Hence, I also agree with L 419.*

Many thanks for your encouraging comment. To address the shortcoming of using model average, the subsequent analyses have therefore used the multi-model min/max/average of trends to communicate the results. We also discussed in our manuscript that "ensemble averages should not be used as a sole ground to infer changes in floods, as this may undermine the actual magnitude of simulated trends" (Line 291).

Considering the key objective of this study (i.e. to compare GHMs capacity in simulating floods and the uncertainty in projected trends) and the complexity of the manuscript in its current form (also noted by the reviewer), we propose to not focus on this aspect in the revision. However, we will make this methodological choice and associated rationale more prominent in the revision.

*L281: is this really 'the spatial pattern of trends' that is evaluated or is it a cell by cell comparison? Please elaborate and have in mind that although a correlation is it can still mean that the overall spatial pattern (i.e. approximate location of increasing and decreasing trends) might still be correct.*

We assume that the reviewer means that "the overall spatial pattern of increasing and decreasing regions might still be correct even when the correlation value is low". During our investigation, we have conducted some visual inspections which confirmed that a low correlation value usually reflect the inconsistency in the spatial pattern between two specific set of trends (an example was provided in the Supplementary). This metric was also used extensively in the climate literature (Kumar et al., 2013;Kiktev et al., 2003;Kiktev et al., 2007) to assess the spatial consistency of trends introduced by different gridded products.

*L 370-384: The authors mention 'a significant difference between trend characteristics from all model grid cells compared to those obtained from the observation locations' and the conclude that '' that trends exhibited from observation locations are not a representative sample of trends obtained from all simulation grid cells'' (L379-380) And then call ''to improve data accessibility and expand streamflow*

*observational networks''. However, if there are such ''significant difference even in data rich regions, how can one justify expanding the network based on the previous finding? Instead to me this reasoning would rather require the need to improve our models instead (notwithstanding the fact that I agree with the data needs mentioned by the authors.)*

We thank the reviewer for noting this out. We will carefully revise our discussion to incorporate this suggestion. Potential changes are (i) to elaborate more on model performance in data-rich regions, and (ii) highlight the need for improved capacity of GHMs in reproducing trends at the Conclusion.

*L 460: Maye the authors can elaborate a little more what an 'flexible adaptation strategy' entails in terms of flood mitigation. Any suggestion on how this can be achieved under tight budgets. Can we as scientists not provide any guidance than just saying 'stay flexible' to those who have to take decisions know?*

We will consider extending our discussion to include feasible strategies and guidance to address high uncertainty in projections of changes in flood hazards.

*L531 & 534: Along the lines of improved GHM: It is not only important that the spatial patterns are being reproduced correctly but also that the timing of the high-flows/floods are being modeled correctly. I.e. 'the flood seasonality patterns can be used as ´ an additional metric to test large-scale hydro-logical models for their ability to reproduce the spatial and temporal flood characteristics.' (Hall and Bloschl, 2018, HESS). ´ As this would give more confidence that the models actually get the flood generation processes correctly.*

We thank the reviewer for this constructive comment. We agree that the timing of flood is a useful metric. This statistic should also be considered in the assessment of model capacity in terms of reproducing flood characteristics at the global and continental scale. We will extend our conclusion and include some corner-stone references (Hall and Blöschl, 2018;Blöschl et al., 2017;Dettinger and Diaz, 2000) to incorporate your suggestion.

*L 538: What does 'constraining ' entail? Please briefly elaborate. Would this prevent the model to adjust to changes in the flood generating processes, as one would expect to happen in some regions of the world. E.g. from snow-melt floods to rainfallgenerated floods?*

This term (i.e. "constraining") refers to the process of using observations to constrain multi-model projections and is commonly used in the climate literature (Padrón et al., 2017;Allen and Ingram, 2002). The purpose of this process is to prevent climate models projecting an unrealistic state of the future climate system (Flato et al., 2013). The constraints are usually the global average values of variables that model developers judge to be important (e.g. the global mean top of the atmosphere energy balance, cloud feedbacks). From our understanding, this process will not violate the fundamental physical processes of the hydrological cycle. We will clarify this terminology in the revision.

*L 550-559: I agree with this call, as this is very important. However, one needs to keep in mind that in many countries maintaining monitoring networks and data curation is/is considered too expensive. Hence it needs to be made clear to decision makers that such data is of importance. However, I know of cases where countries/agencies have been or are currently considering discontinuing their data networks, as they don't see the benefit or don't see their data being used (partly lack of proper citation of the (often freely available) original data source). This implication needs to be kept in mind when large*

*datasets of observational data are being compiled and subsequently only credit is given to the compiled data. . . This hides to the funding/responsible agencies the usage of their data (i.e. the original data source) and might lead to the misconception that their data is not being needed/downloaded and hence the data network can be discontinued and to allocate funds to more (perceived) useful sectors. . .*

We thank the reviewer for the comment. We agree this is very important to make national data authorities aware of the importance of their works. We will specifically emphasize the role of data "end-user" in making streamflow data more FAIR by properly acknowledging the efforts and merits that data providers deserve.

*Fig S5: Suggest using same y-axis scale for all panels on the left/right to be able to compare the regions better with each another.*

We will revise the figure in our revision to ensure a consistent scale on the y-axis is used.

**Reference**

Allen, M. R., and Ingram, W. J.: Constraints on future changes in climate and the hydrologic cycle, Nature, 419, 224-232, 2002.

Blöschl, G., Hall, J., Parajka, J., Perdigão, R. A. P., Merz, B., Arheimer, B., Aronica, G. T., Bilibashi, A., Bonacci, O., Borga, M., Čanjevac, I., Castellarin, A., Chirico, G. B., Claps, P., Fiala, K., Frolova, N., Gorbachova, L., Gül, A., Hannaford, J., Harrigan, S., Kireeva, M., Kiss, A., Kjeldsen, T. R., Kohnová, S., Koskela, J. J., Ledvinka, O., Macdonald, N., Mavrova-Guirguinova, M., Mediero, L., Merz, R., Molnar, P., Montanari, A., Murphy, C., Osuch, M., Ovcharuk, V., Radevski, I., Rogger, M., Salinas, J. L., Sauquet, E., Šraj, M., Szolgay, J., Viglione, A., Volpi, E., Wilson, D., Zaimi, K., and Živković, N.: Changing climate shifts timing of European floods, Science, 357, 588, 2017.

Dettinger, M. D., and Diaz, H. F.: Global Characteristics of Stream Flow Seasonality and Variability, Journal of Hydrometeorology, 1, 289-310, 10.1175/1525-7541(2000)001<0289:GCOSFS>2.0.CO;2, 2000.

Do, H. X., Westra, S., and Michael, L.: A global-scale investigation of trends in annual maximum streamflow, Journal of Hydrology, 10.1016/j.jhydrol.2017.06.015, 2017.

Flato, G., Marotzke, J., Abiodun, B., Braconnot, P., Chou, S. C., Collins, W., Cox, P., Driouech, F., Emori, S., Eyring, V., Forest, C., Gleckler, P., Guilyardi, E., Jakob, C., Kattsov, V., Reason, C., and Rummukainen, M.: Evaluation of climate models, in: Climate change 2013: The physical science basis. Contribution of Working Group I to the Fifth Assessment Report of the Intergovernmental Panel on Climate Change edited by: Stocker, T. F., Cambridge University Press, Cambridge, United Kingdom and New York, NY, 741-866, 2013.

Galton, F.: Regression towards mediocrity in hereditary stature, The Journal of the Anthropological Institute of Great Britain and Ireland, 15, 246-263, 1886.

Hall, J., and Blöschl, G.: Spatial patterns and characteristics of flood seasonality in Europe, Hydrol. Earth Syst. Sci., 22, 3883-3901, 10.5194/hess-22-3883-2018, 2018.

Katragkou, E., García Díez, M., Vautard, R., Sobolowski, S. P., Zanis, P., Alexandri, G., Cardoso, R. M., Colette, A., Fernández Fernández, J., and Gobiet, A.: Regional climate hindcast simulations within EURO-CORDEX: evaluation of a WRF multi-physics ensemble, 2015.

Kiktev, D., Sexton, D. M., Alexander, L., and Folland, C. K.: Comparison of modeled and observed trends in indices of daily climate extremes, Journal of Climate, 16, 3560-3571, 2003.

Kiktev, D., Caesar, J., Alexander, L. V., Shiogama, H., and Collier, M.: Comparison of observed and multimodeled trends in annual extremes of temperature and precipitation, Geophysical research letters, 34, 2007.

Kron, W.: Flood Risk = Hazard • Values • Vulnerability, Water International, 30, 58-68, 10.1080/02508060508691837, 2005.

Kumar, S., Merwade, V., Kinter III, J. L., and Niyogi, D.: Evaluation of temperature and precipitation trends and long-term persistence in CMIP5 twentieth-century climate simulations, Journal of Climate, 26, 4168-4185, 2013.

Mallakpour, I., and Villarini, G.: The changing nature of flooding across the central United States, Nature Clim. Change, 5, 250-254, 10.1038/nclimate2516, 2015.

Masaki, Y., Hanasaki, N., Biemans, H., Schmied, H. M., Tang, Q., Wada, Y., Gosling, S. N., Takahashi, K., and Hijioka, Y.: Intercomparison of global river discharge simulations focusing on dam operation— multiple models analysis in two case-study river basins, Missouri–Mississippi and Green–Colorado, Environmental Research Letters, 12, 055002, 2017.

Padrón, R. S., Gudmundsson, L., Greve, P., and Seneviratne, S. I.: Large-Scale Controls of the Surface Water Balance Over Land: Insights From a Systematic Review and Meta-Analysis, Water Resources Research, 53, 9659-9678, doi:10.1002/2017WR021215, 2017.

---

## Author Response (AR1)

**Response to reviewers**

**"Historical and future changes in global flood magnitude – evidence from a modelobservation investigation" [HESS-2019-388]**

This document provides response to Editor and Reviewers for the above manuscript.

*The comments from Editor/Reviewers are quoted in blue, italic text.* Previous responses from the online discussion are formatted in normal text. New responses after revision are in red.

**Response to Editor**

*Editor Decision: Publish subject to revisions (further review by editor and referees) (03 Nov 2019) by Louise Slater*

**Dear Authors,**

Thank you for the detailed responses to the two reviews on your paper.

Overall, both reviewers were quite positive about the study, and I agree with many of their comments. They made some valuable suggestions regarding important topics such as: identifying the research gap and clarifying the motivation; providing an outline (or schematic) of the methodology; attempting to relate the results to the model characteristics (where feasible); clarifying the terminology to avoid misinterpretation; providing further discussion of results and interpretation of findings; and making some of the data available for reproducibility.

*I find your proposed modifications appropriate and would like to invite you to go ahead and submit a revised manuscript, which will be returned to the reviewers for further review.*

Sincerely,

**Louise Slater**

We would like to thank the Editor for her encouraging evaluation of our manuscript. Our revision has incorporated the suggestions from Reviewers, including:

- Substantial revisions of the introduction to better highlight the research motivation.
- A new figure has been added to Methodology section to summarise the research outline.
- Significant changes across the manuscript to ensure the results are presented/interpreted in a concise manner, and avoid potential misinterpretation. Major changes are: (i) new material in supplementary to better highlight the difference across models; (ii) significant revision of Section 2 and Section 3 to communicate the results more precisely; (iii) adding clarifications when needed; and (iv) restructuring Section 4 to make the key findings more appealing.
- We also provided csv file contains historical trends for MAX7 index across 3666 locations derived from GSIM dataset and all model simulations.

We would like to draw the Editor attention that the manuscript has an additional author (Camelia-Eliza Telteu), and has corrected the name order of another author (Julien Eric Stanislas Boulange). Camelia has synthesized the key differences and similarities across GHMs (supplementary Section 1), and contributed substantially in the revision to improve the quality of this manuscript.

**Response to Reviewer #1**

We would like to thank Reviewer #1 for the constructive comments which will help to improve the quality of our manuscript. Each of the reviewer's comments is quoted in *blue, italic text*, followed by our reply formatted in normal text. Additional responses (after online discussion) after revisions are in red.

**General comments**

Not content related: the authors list reads like a "manel". Sad to see so little inclusion of other genders than male. Maybe something to think about for future studies?

We thanks the reviewer for their note about the broader diversity in the author panel in such global scale initiative. We will aim for improved inclusion of gender diversity in future investigations.

I do miss the overarching research motivation of this study. Even though the introduction contains a good amount of background literature and describes the three research objectives set, it does not become clear what the research gap is that the present study intends to fill. Or in short, how are the objectives derived and what is the (societal) relevance of the study? Please ensure this is clearly and concisely explained in the revised version of the manuscript.

Thank you for your note that the research motivation should be presented more prominently. We will revise the introduction and consider revisions focusing on: (i) highlighting the contribution of this study to address existing research gap in the field, (ii) including the motivation from the perspective of global hydrological model users (e.g. decision makers).

We have revised the Introduction substantially to make our motivation clearer. The objectives have also been rewritten into bullet points for improved readability.

The models used in the study differ in quite some characteristics/schematization. An overview of these differences would be very useful (also possible as supplement). Besides, none of the deviations between results obtained with different GHMs is related to those different characteristics/schematization. I would guess that they do play a big role in explaining the obtained results and thus I recommend extending the manuscript with such an analysis (or a clear and convincing statement why not).

We appreciate the reviewer's suggestion about exploring the role of model conceptualization in simulating trends in floods. We will explore the possibility of having a supplementary table outlining the differences among GHMs in the revision.

**A more detailed description of GHMs as well as key difference across models (and between versions of the same model) have been added in Supplementary.**

We note, however, that a detailed documentation of differences in model characteristics does not necessarily enable evidence of why the models produce different outputs. From our understanding, previous studies have related the impacts on peak flows of some specific processes such as routing scheme or reservoir algorithms (Zhao et al., 2017; Masaki et al., 2017), but it does not mean that a similar impact is presented in simulated trends (also highlighted in our manuscript at Line 85). This is also a motivation of this study, as we want to compare the trends of a high flow indicator simulated by different GHMs.

To nevertheless incorporate the reviewer's comment, we will make the main objective of this study (i.e. a comparison of model capacity in simulating trends in a flood indicator and an assessment of uncertainty of projected changes in floods) more prominent in the introduction. We will also mention the reviewer's suggestion (i.e. to assess the impact of model schematization on changes in flood hazard) in the conclusion as a potential research direction.

The Introduction has been revised to better communicate key motivations and objectives of the study. We also added, in the Conclusion, a potential research direction to explore the reasons leading to discrepancies in trends simulated by different models.

The manuscript reads bit lengthy sometimes. To some extent, this is the result of reporting a lot of numbers, which are partially also provided within tables. My recommendation would be to let the tables and figures speak for themselves and to check which reported numbers can be neglected as they are not directly needed for understanding the methodology or results. Plus a check whether more concise wording could be used.

The manuscript will be carefully revisited to focus more on the key findings, reduce any redundant information and present the results in a more concise manner. We will focus especially on Section 3.1 where we will remove any descriptions that have already available in the tables or figures.

We have revised the manuscript substantially to reduce redundancy, and to improve readability wherever possible (the vast majority of these changes are in Section 3). We also reorganized Section 4 to improve readability and better highlight the key findings.

This study is very much focussed on data and their analysis. Extending the implication of the findings of the study to the societal dimension would greatly benefit the manuscript to make the results more tangible and applicable.

We thank the reviewer for the suggestions on extending the study findings to societal dimension. We note that the key motivation of this study is to explore the level of consistency of trends detected from streamflow observations and model simulations, which is currently underrepresented in the literature. As a result, we would like to focus on this research pathway and will refine the introduction to highlight this objective better.

The Introduction is now revised substantially to better communicate our research pathway (to explore the uncertainty of model-based inferences on changes in floods).

I would very much welcome it if at least the data pairs for the observation stations are provided via a supplement. This would be in the spirit of FAIR hydrologic modelling, thus increasing reproducibility of your findings. We will upload the observed and modelled trends at each station as a supplementary (csv files) together with the revised manuscript.

The csv file containing geographical coordinates and historical trends (calculated from all datasets) across 3,666 locations has been uploaded as Supplementary.

**Specific comments**

• P1/L29+L30: An agreement of 12-25 % can hardly be named "moderate agreement", can it?

The abstract will be revised to ensure appropriate terminologies are used, potentially by changing "moderate" to "low-to-moderate".

• P1/L31 "significant differences": please specify what kind of differences you refer to.

We will clarify that the characteristics of trends (trend mean, trend standard deviation) simulated by GHMs forced with historical climate is significantly different to that simulated by GHMs forced by bias corrected climate model output.

The abstract was revised as outlined in the two responses above.

**• P2/L52 "specific regions": specific regions such as? Please specify.**

We will clarify that this the statement may only applicable where rainfall plays the dominant role in flood occurrence.

Clarification added.

**• P2/L53 "recent evidence": what evidence? Please provide name, source, etc.**

The following sentences (Lines 54-56) in fact have extended our discussion and provided some evidence for this statement. We noted that this may be unclear and will revisit this paragraph to ensure the statement is justified.

**We have added two references to justify this statement.**

• P2/L52-L66: what about the role humans play in changing flood hazard? Please add this dimension to the paragraph.

We will add the impact of human activities to changes in flood hazard at the end of this paragraph.

Impact of human activities added.

• P3/L77 "factorial experiments": what are you referring to with this term? Please explain or use more common terminology.

"Factorial experiments" indicate studies analysing the effect of different factors (e.g. land use change) on the response variable (e.g. changes in floods), as well as the effects of interactions among the factors on the response variable. In the context of hydrological modelling, the impact of atmospheric forcing, land use change and other drivers of change on streamflow trends could be "turn on/off" to provide a full "factorial experiment design". We will rewrite this statement to improve readability in the revision.

The statement was rewritten and "factorial experiments" is no longer used.

**• P3/L87-L100: The description of the research objectives could profit from using bullet points**

We thank the reviewer for their suggestion. We will consider using bullet points to separate the research objectives.

The Introduction has been revised, and the objectives are now presented in bullet points.

**• P4/L105-L108: why are these models used? Why are some others available within ISIMIP not used? Please clarify.**

There is no model selection in this study. We actually used all GHMs that have provided discharge data within Phase 2a and 2b simulations at the time this study was initiated (June 2018). In the revision, we will highlight this fact in a transparent manner.

We have added clarification of model choice in section 2.1.

• P4/L122-L126: As mentioned in the general comments, a (technical) description of the models is needed. A particular focus should be on how the models changed between ISIMIP2a and ISIMIP2b and whether those changes may have influence results (or not). Possible changes in e.g. functionality, spatial resolution, etc. may have had a great impact on results and thus affecting the comparison performed in your study. I thus strongly disagree that checking this is outside the context of the study.

Similar to our response to the general comment from the reviewer, a precise conclusion about the impact of changes in models (e.g. functionality, spatial resolution) on trends in floods should be based on a full multi-model experiment (i.e. to compare trends simulated by different versions of the same GHM), which is unfortunately not readily available. Although we are aware that changes and bug-fixes done in MPI-HM affect only the human impact simulations (and the influence is insignificantly), it is not straightforward to generalize this conclusion. We will aim for some extended discussion, but would like to keep our statement as-is (i.e. checking the affects is outside the context of the study).

We also note that the issues raised here and in the earlier comment show the need for the next step of model inter-comparisons which should focus on diagnosing the reasons for differences across models. We have mentioned about this need in our manuscript (Lines 542-547) and will consider make this call more prominent in the revision.

We have added a new sub-section in Supplementary (section 3.3) to illustrate how modifications in GHMs could influence the results. As only WaterGAP simulations are available for this analysis (the

impact is minor), it is not possible to draw a common conclusion across all models, we have noted in both the manuscript and Supplementary that the potential effects of technical discrepancies cannot be checked in the context of this study.

**• P6/L144-L147: what was the reason to not only use the un-routed runoff for all catchments? Wouldn't this increase comparability between results as it removes (unnecessary) transformation of results and units?**

The reason is that for large catchments, observed discharge and unrouted runoff are not comparable. In some very large basins, it takes one to three months for upstream runoff to reach river mouth through the channels (and be measured as discharge here by some observing gauges). The same magnitude of basin total runoff, depending on its spatial distribution (i.e., evenly distributed versus concentrated in the downstream), could generate rather different discharge after routing. Therefore, we adopt different procedures for large and small basins to achieve maximum consistency in model-observation comparisons.

• P6/L149 "catchment area": which area estimates did you use? For all models the same? Per model based on catchment delineation? Please clarify to avoid that data was used inconsistently.

We only used the reported catchment area of each stream-gauge in this calculation. We will clarify about this technical aspect in the revision.

Clarification added.

• P7/L177-L185: great you are pointing out the differences in methodology!

Thank you for your encouragement.

• P11/L228-L230: these lines read as if they should not be part of the methodology, rather of the results/discussion section. The fact that that there may be 'hot-spots' of future flood hazard should be discussed in more detail and thus deserves a more prominent location in the manuscript.

We appreciate the reviewer's suggestion. We will revisit this paragraph and consider highlighting the "hot-spots" aspect more in the revision.

Section 2.4.2 has been revised to clarify the methodology. However, we decided to not extend our discussion about "hot-spots" in flood hazards – considering the high uncertainty presented in the GCM-GHM ensemble. From our perspective, the current discussion about "high-risk" locations are under-sampled is appropriate in the context of this investigation.

**• P11/L231 "each grid-cell": each grid-cell or only those paired with a GSIM-location? Does not become very clear from reading.**

This analysis was conducted for each grid-cell across the globe, regardless there is stream-gauge or not. We will clarify about this to avoid confusion.

We revised the description to "each grid-cell available in the discharge simulation grid" for clarity.

**• P11/L235-L237: why was this done? What does it add?**

This step was included to assess whether the locations that robustly projected with increasing/decreasing trends in flood hazard (i.e. the magnitude of MAX7 index increases/decreases significantly during the 2006-2099 period) has been observed adequately by the current streamflow observation system.

We will revisit this section in the revision to improve clarity.

Section 2.4.2 has been revised substantially for improved clarity.

• P11/L240-L247: You describe the observation and simulations, but you do not mention the reasons behind it. Why are certain areas experiencing increases and others decreases? Is it all hydrology or not? Can we say something about the driving factors behind it? Please add.

We acknowledge the reviewer's perspective about the importance of attributing changes in flood hazards to hydrological or climatic mechanisms factors. We noted, however, that the key objective of the present study is to assess model capacity rather than exploring the mechanisms driving changes in flood hazard. The reviewer also noted that the manuscript has been quite complex in its current state already. As a result, we propose to not include these discussions in the revision. Instead, we will make our objectives clearer, and will clarify that the paper does not focus on explaining the mechanisms driving changes in floods.

The Introduction was revised to further highlight the motivation and objectives of our investigation.

**• P12/L255+L256: What is the implication of this finding?**

We discussed about the implication of this finding at line 290, in which we suggested that averaging will reduce the magnitude of trends and thus ensemble average should not be used as a sole ground to infer change in floods. This is also a motivation for us to provide the range of trend characteristics across all ensemble members.

We will revisit our discussion to communicate this implication clearer.

P12/L261+L262: So, if it is not visible through Figure 2, how can it be an alternative explanation? This sounds contradictory to me – either it's possible based on your results or your results say it's not a thing. Please clarify.

The intention of this statement was to indicate that we would explore GHM's performance in more detail (i.e. through the next paragraphs/sections) because Figure 2 alone was insufficient to explain such feature.

We found this statement may be confusing and will revise it to improve clarity.

Section 3.1 has been revised substantially to incorporate the two suggestions above from the reviewer.

**Figure 2: A bit bigger figure (maybe with subplots of USA, EU) would help seeing the differences between differences between historical trends.**

We note that the Supplementary has included sub-region plots for this figure. This figure is also useful to highlight the "white spaces" over many regions, which was then linked to our call for more streamflow observation.

We will explore the options to improve graphical quality of this figure in the revision. Some possible options are to include the vector graphic or use another colour pallet.

Figure 2 has been revised, but the focus was not on highlighting trends in sub-regions. Instead, we added scatter-plots to highlight the difference in simulated trends of GSWP3 and GCMHIND. The figure will be provided as vector graphic to ensure high quality of the final image.

**• P14/L285-L287 and Table 3: what are possible reasons for the different model results? Model structure, processes simulated per model, spatial resolution, routing schemes applied or something else? Would be great if you could elaborate a bit on this.**

From our perspective, this question is not straightforward to answer due to the number of participating models (six) and the many factors involved (e.g. the individual and collective effects of differences in model conceptualization, spatial resolution and routing scheme). These aspects (i.e. possible reasons for different trends simulated by different GHMs) in fact is s till under-represented in the literature. Even when model differences are documented extensively, it is still challenging to precisely attribute output discrepancies to a specific (or a set of) factor(s) without supports from another set of GHM simulations (e.g. checking the sensitivity of simulated trends corresponding to changes in a specific factor). Nevertheless, we will consider to elaborate about potential sources of differences in model outputs in the revision. We will also highlight this in the conclusion as a potential research pathway.

We have expanded our supplementary to cover the key features of different models (supplementary Section 1). As the impact of model structure differences to outputs cannot be explicitly identified (supplementary section 1.2), we emphasized in Conclusion section the call for more investigations to explore the reasons of output discrepancies (despite having a common climate forcing as input).

**• P14/L290+L291: if it should not be used as "sole ground", what other measures would you (like to) use to infer changes in floods?**

We will clarify in the revision that the range of all ensemble members should be used to illustrate the spread of simulated trends (e.g. the information showed in Table 4).

We have revised this section to provide a better narrative for these ideas.

• P23/L463+L464: Does that mean your results are not usable to help inform flood management practices in less well-observed areas? What would be the implications? Please elaborate briefly on the consequences of your results.

The intention of this statement is to set the stage for our next analysis which shows the regions projected with increasing flood hazards are under-sampled, and ultimately leads to our call for more attention to improved streamflow observations. We will revise our discussion to improve the narrative of these ideas.

Section 3.3 has been revised substantially to improve clarity.

• P24/L475-L478: What would be ways forward to reduce the dependency on not evenly spatially distributed observation systems? Which opportunities do, for instance, remotely sensed data products bring? Please put your findings into context, here and/or the conclusions section.

This statement was used as a ground for our call (at the conclusion) for more FAIR streamflow observations to support hydrological research. We acknowledge that the narrative may need improvement and will revisit the paper, potentially including some of the reviewer's suggestions.

We have added a call for using remotely sensed data products and runoff reanalysis to offset observation scarcity at the end of this section.

P26/L533-L537: It should be added that also the routing schemes of GHMs should improve, not only the runoff. Well timed runoff with right magnitude can still result if inaccurate streamflow if the routing scheme is too simplistic. Vice versa, higher-order routing schemes cannot perform at their best if input runoff is not accurate. Relevant literature: Hoch et al., 2019 (https://doi.org/10.5194/nhess-19-1723-2019) and Zhao et al., 2017 (https://doi.org/10.1088/1748-9326/aa7250).

We will extend our discussion to include the importance of the routing scheme on GHMs' performance and the need to improve this important feature in future GHM generations.

Our conclusion has been extended to incorporate the reviewer's comment.

**P27/L550-L559: I very much agree with this, well written!**

We thank the reviewer for their encouraging comment.

**Response to Reviewer #2**

We would like to thank Reviewer #2 for the constructive comments that will help us to improve the quality of the manuscript. For clarity, we formatted reviewer's comments in *blue, italic text*, while our responses are formatted in normal text. Additional responses after revisions are in red.

This manuscript describes the work of an extensive model study and in its final version will be for sure appreciated by the readers of HESS.

We thank the reviewer for the encouraging evaluation.

General Comments: As the study presented is quite extensive, it is sometimes difficult to follow study setup and all the analysis steps. Therefore, the authors should provide a detailed schematic, showing the main building blocks of their study and the different steps of analysis (preferably showing the section numbers in the schematic as well) to allow the reader to have a complete 'picture' of the study design, before embarking on the details in the main text.

We will include an additional figure at the start of Section 2 to provide a complete picture of the analyses presented in this study.

New figure added.

Additionally, due to the complexity of the study and details provided in the result section, I think a summary table or bullet points at the end of the study would be helpful for the reader to get a better overview of the key results obtained.

Thank you for pointing out this issue from the reader's perspective. In our revision, we will carefully revisit the manuscript to improve readability and simplify the contents where relevant. We will also consider your suggestions to format the main findings in bullet points to communicate the key points better.

We have revised Section 4 substantially to improved clarity and readability.

Another important point is that the study uses 7-day annual maximum as a surrogate for 'food'. This fact needs to be made more explicit throughout the study to avoid misunderstandings from the general perception of flood, which would shorter (e.g. often 1-day). This is of importance, as the results might be quite different. I.e. a single day peak value trend study will show different results, not only in terms of magnitude of change, but also in terms of the flood hydrograph shape. E.g. if floods would become flashier in some location in future, it might look as if the trend of a 7-day maximum might not change at all or get smaller, but the peak day could be of much higher magnitude. The authors need to make sure they call the variable under investigation for what it is, i.e. not calling it 'flood', 'peak discharge' or 'streamflow maximum' to avoid misunderstanding of the results.

We thank the reviewer for the suggestion. We would like to note that in our preliminary analysis, we also analysed the annual maximum values of daily streamflow (i.e. 1-day flood time series, MAX index).

Although trend at specific site may vary between these two indices, we found that the key conclusions (e.g. the regional pattern of increasing/decreasing trends, the consistency between trends exhibited from observed data and that obtained from simulated data) are quite similar, regardless which index being used (i.e. MAX or MAX7 index). To address your concerns, we will revise our manuscript substantially to ensure: (i) there is sufficient information about the consistency between results introduced by MAX and MAX7 index, and (ii) the terminologies are used appropriately (e.g. replace "streamflow maximum" by "7-day streamflow maximum").

We have carefully revisited the manuscript to avoid misinterpretation of our findings.

**Along this line, I also think that the title '... changes in global flood magnitude ... ' is also misleading. The study shows rather an 'global assessment of the 7-day annual maximum average value'. Please consider changing the title to better represent the content of the study.**

We acknowledge the reviewer's concern and will consider a change in the title. A possible option is to replace "magnitude" by "indicator" (i.e. "Historical and future changes in an indicator of global flood hazard - evidence from a model-observation investigation"), due to the fact that MAX7 can also be used as an indicator for floods, and using MAX index has generally led to comparable results to that of MAX7 index (discussed above). From our perspective, this proposed title is not misleading, and can potentially reach a broader readership than using a too technical title such as "Global changes in 7-day annual maximum average value".

In line with our response above, the abstract, introduction and conclusion have been revised to emphasize that the findings were based on analysing MAX7 index. As MAX7 index can also be used as a proxy of flood magnitude, we proposed to keep the title as-is (i.e., "Historical and future changes in global flood magnitude - evidence from a model-observation investigation") to attract the broad readership of HESS.

Additionally, to avoid misinterpretations of your results please avoid using the term 'hazard' in its current form in the manuscript, as hazard means: hazard=risk\*exposure (which is not the correct terminology here). The same also applies to the term 'risk' which is related to 'probability and consequences'.

We noted that this study was developed from the perspective that 'hazard' (e.g. flood magnitude, frequency or inundation) is a component of 'risk' (i.e. Flood Risk = Hazard x Values x Vulnerability; Kron, 2005). From this point of view, we judge 'hazard' the appropriate terminology to refer to the MAX7 streamflow index. We acknowledge the reviewer's concern about the use of the term 'risk' and will carefully evaluate the manuscript and clarify/make changes where relevant to ensure the appropriateness of each terminology.

We have clarified our definition of "risk" at the Methodology, Results, and Conclusion sections to avoid misinterpretations.

In this manuscript I feel that the GHM are used by the authors as 'black-box' that give some output. However, for this study to be valuable, it would be important that the authors would try to relate the observed differences/deviations in the outputs to the actual differences in the hydrological model setup.

**The authors just state "... there are potential effects of technical discrepancies to the findings which cannot be checked in the context of this study" (L 126).**

We agree with the reviewer that the relationship between the model's structure and model's capacity in simulating trends in floods is an important aspect. However, addressing this comment is not straightforward, as there are a total of six models with many factors (e.g. routing schemes, spatial resolution, and parameterisations) that could individually or collectively lead to output discrepancies. These aspects (i.e. possible reasons for different trends simulated by different GHMs) in fact is still under-represented in the literature. From our perspective, this type of investigation deserves a separate paper by itself as the work involved should be tremendous, and potentially involved another set of simulations (e.g. to check the sensitivity of simulated trends corresponding to changes in a specific factor).

In the revision, we will refine the introduction to clarify the key objectives of our study, which is to compare trends observed by different models and the uncertainty in projected trends rather than to explore the mechanisms driving discrepancies in model outputs. We will also highlight the reviewer's comment in the conclusion as a potential research direction.

To address the reviewer's concerns, we have provided a summary of model characteristics and mention the impacts of technical discrepancies wherever possible (section 2.1, supplementary section 1.2, and supplementary section 3.3). As an explicit statement of technical discrepancy impacts on simulated trends was not available through our investigation (also discussed in our responses to Reviewer#1), we have included a call (in the Conclusion) for future research to explore the reasons behind this feature.

However, I think based on the model selection, the authors should have a notion of why they selected certain models and what the key differences are. Hence, the authors should at least try to come up (also based on past literature) with some sort of reasoning for model selection and also more importantly an interpretation of their findings...

In this study, we did not make any model selections. Specifically, we used all hydrological models that have produced discharge outputs for both Phase 2a and 2b at the time this study was initiated (June 2018). In the revision, we will highlight this fact better to avoid confusion.

We have clarified model choices in the revision.

**For example, are the changes the models are giving as an output considered in line with the current understanding of the effects of climate change on floods or are there surprising results? I think this could be done in a separate paragraph discussing/comparing with previous literature.**

Our manuscript has highlighted that the historical trends obtained in the present study are consistent with what has been reported in the literature (Lines 240-247). The reviewer suggested that simulated trends should also be linked to the current understanding of the effects of climate change. Although this aspect is important, we intend to not cover it as our objectives are not to attribute change in flood hazard to climate change or human activities. For historical trends (1971-2005; or ISIMIP2a), the focus was to compare model capacity in reproducing observed trends and compare the performance of simulations driven by observed (GSWP3 simulations) and modelled atmospheric forcing (GCMHIND simulations). The ultimate goal is to show the uncertainty of trends in the MAX7 index detected from

the current GHM-GCM ensemble. For future trends (2006-2099), the focus was on the robustness of projected trends introduced by the ensemble members.

In addition, there is another ISIMIP investigation dedicating on river flow changes attribution, and thus we decided to exclude this aspect from this manuscript to avoid overlap.

To further incorporate the reviewer's suggestion, we have reorganize the final section (Summary and conclusion) to emphasize that the results of our findings are consistent to previous studies (the first point of the summary).

In several instances in the manuscript, the authors are highlighting the 'substantial influence of the atmospheric forcing in driving the spatial structure of the simulated trend'. I think this is another important point that needs to be discussed in more detail in the discussion section, i.e. why to the hydrological models have little influence...

The hindcast simulations of the global climate model are forced by historical  $CO_2$  (Katragkou et al., 2015), and so the timing of wet/dry periods or the spatial distribution of precipitation will be different from what has been observed in the past. As precipitation is arguably one of the most important inputs for streamflow simulation, it is expectable that GCMHIND trends will have a more prominent impact on the spatial patterns of simulated trends relative to model structure. We will consider including this justification in the revision.

We have added the above discussion at the end of Section 3.1 to clarify that GCM outputs generally have lower capacity to simulate the spatial structure of weather extremes, thus the lower capacity of GCMHIND trends in MAX7 index is somewhat expected.

Overall, I think a new separate discussion section of the results of such a complex analysis would be beneficial, as this would free up the room for a better refined summary and conclusion section, that focused on the key results and the overall implications of the results not just for the scientific world but also for the 'end-users', such as decision makers etc.

Thank you for suggesting this potential improvement. We will revisit the whole paper to better discuss the findings and improve the paper readability. Some opportunities for improvement have been identified, which we believe will help the paper streamlined better:

- Revisit our introduction to clearly state the research objectives and narrate the analyses.

- Include an additional figure (in line with our previous response) to show the overall framework of the study and how does it address the research questions.

- Simplify the contents where relevant, potentially in Section 3.1, to exclude redundant information and make the analyses more focus.

We will also consider your suggestion (i.e. having a separate discussion section) during our revision.

In the revision, we have revised the manuscript and supplementary substantially to make the key findings better come through. We note, however, that we decided to keep the headings as-is.

**Specific Comments:**

**L37: For clarity, please provide significance level used in this study in parentheses.**

We will add the level of significance (10% two-sided) in the revision for clarity.

**L38: replace the term 'high-risk location'.**

Thanks for noting, we will evaluate the terminologies across the manuscript, potentially using "locations robustly projected with increasing flood hazards".

The abstract has been revised to address the two concerns above from the reviewer.

**L54: Please provide reverence to this statement**

The following sentences (Lines 54-62) in fact has extended our discussion and provided some evidence and references for this statement. We noted that this may be unclear and will revisit this paragraph to ensure the statement is justified.

We have added two references to justify our statement.

**L77: What is 'factorial evidence' in this regard? Please elaborate.**

"Factorial experiments" refer to studies analysing the effect of different factors (e.g. land use change) on the response variable (e.g. changes in floods), as well as the effects of the interactions among the factors on the response variable. In the context of hydrological modelling, the impact of atmospheric forcing, land use change and human water management on streamflow trends could be "turn on/off" to provide a full "factorial experiment design". In the revision, we will revise this statement to improve clarity.

**We have revised this statement.**

**L121-122: Please elaborate why the authors think that the 'naturalised runs and the human impact runs exhibit similar characteristics of trend' Would one not expect considerable differences?**

We thank the reviewer for the suggestion, which will be incorporated in our revision. Some potential reasons are the spatial distribution of stream gauges, which may be biased toward regions with insignificant changes in human intervention within the reference period (1971-2005), or the inclusions of small catchments (more that 3000 catchments with area less than 9000km2), and floods are more sensitive to changes in extreme precipitation relative to the accumulated basin-wide influence of human impacts.

This paragraph has been revised (the above response was added) to incorporate the reviewer's comment.

L126: What are the 'potential effects'. Can you briefly elaborate.

The most pronounced effect comes from the difference in the versions of GHMs that were used in ISIMIP2a and ISIMIP2b. Specifically, ISIMIP2a was designed as an evaluation framework to improve the models for the projection phase isimip2b. As a result, the assessment using historical simulation (from 1971-2005) may not reflect the "true" model capacity in simulating trends in floods during the future period (2006-2099). We will elaborate on this fact and the potential effects of different model versions in the revision. However, as mentioned in our response to Reviewer#1, a solid conclusion about these effects may not be available.

In line with our response to Reviewer#1, we have extended our discussions, and included additional information in supplementary to elaborate on the effects of technical discrepancies.

**L127: Please also elaborate what the effects/impacts of this on the results are.**

Thanks for your comment. We will revise the manuscript to elaborate on the potential effects/impacts of technical differences across GHMs, potentially including:

- Different drainage direction maps across different models could lead to gauging stations (in some rare cases) that do not lie on the river network (Masaki et al., 2017).

- Different models do not have the same set of coastal cells which may lead to some minor effect to the statistics when averaged across all simulation grid-cells.

- ORCHIDEE runs on 1-degree resolution but is routed at 0.5-degree resolution and thus influenced by a stronger spatial averaging that could lead to more flatten discharge time series.

We have revised this statement to elaborate the impacts of differences in the number of coastal gridcells.

**L158: What is the rationale of 335 days. Please explain briefly.**

The rationale of this choice is every single year must have at least 90% of streamflow data available. This criterion is a common data filtering condition in large-scale observation-based investigation (Do et al., 2017;Mallakpour and Villarini, 2015). This data criterion was chosen to fit the purpose of a hybrid observation-simulation study. We will consider clarify this methodological choice in the revision.

We have added a brief explanation for this methodological choice.

**L172: Fig1: These colours are not 'safe' for colour-blind readers. Please use different colour combination**

We will revise this figure in the revision to address this concern. Specifically, we will consider the use of an eight-color discrete palette that is colorblind safe (available in ggthemes R package at <a href="https://rdrr.io/cran/ggthemes/man/colorblind.html">https://rdrr.io/cran/ggthemes/man/colorblind.html</a>).

This figure has been revised.

**L184: 'Our preliminary analysis. . . did not lead to substantial changes'. So what were the 'not so substantial changes' one is wondering?**

The preliminary assessment showed that the regional patterns of changes detected from MAX and MAX7 indices are generally consistent. We will clarify this point in our revision.

Clarification added.

**L192: Can you please name the 'three identified objectives' again as it is quite difficult to keep up with this extensive work.**

We will consider provide the identified objectives as bullet points to remind readers about the focus of this study.

We have revised this statement to incorporate the reviewer's suggestion.

**L210: To spare the reader from having to go to the original reference, please name the field significance test used and elaborate briefly what exactly is evaluated.**

We will incorporate your suggestions in the revision by adding a brief explanation about the bootstrapping technique that was used.

We have added a note that the technique is briefly explained in Table 2.

**L211: What 'Pearson's (spatial) correlation' was used? Reference? What variables are correlated?**

Here we computed the Pearson's correlation *r* metric (Kiktev et al., 2003;Galton, 1886) to represent the spatial consistency between two sets of trends in MAX7 index. We will clarify this statistical technique to improve clarity in the revision.

Clarification added.

**L220: Please replace the term 'flood hazard' with something more appropriate to what has been done. This also applies to the subsequent usage, as well as the term 'floodrisk' later used in the manuscript.**

As mentioned in our previous response, we think flood hazard is the appropriate term to refer to the magnitude of MAX7 index. Nevertheless, we will carefully evaluate the manuscript to ensure the most appropriate terminologies are used.

In line with our response to the reviewer's general comments, we have revisited the manuscript and clarify our definition of "risk" in the revision.

L245 & 493:to me it does not look like norther Europe has increasing trends. Scandinavia etc looks decreasing. . . . Please check.

We thank the reviewer for noting out this mistake – which should be "the northern part of Western Europe". We will revise the manuscript to ensure correct description is presented.

We have fixed this mistake.

L258: I agree, very much with this point. The study analyses 'extremes (i.e. floods) but then model 'averages' are provided. His is counter intuitive. This can lead to strong underestimation of the actual changes. The usage of averages vs individual models that show extremes should be better discussed in the discussion section. Hence, I also agree with L 419.

Many thanks for your encouraging comment. To address the shortcoming of using model average, the subsequent analyses have therefore used the multi-model min/max/average of trends to communicate the results. We also discussed in our manuscript that "ensemble averages should not be used as a sole ground to infer changes in floods, as this may undermine the actual magnitude of simulated trends" (Line 291).

Considering the key objective of this study (i.e. to compare GHMs capacity in simulating floods and the uncertainty in projected trends) and the complexity of the manuscript in its current form (also noted by the reviewer), we propose to not focus on this aspect in the revision. However, we will make this methodological choice and associated rationale more prominent in the revision.

To incorporate the reviewer's suggestion, we have revised section 3.1 to highlight the fact that multi model average may potentially mask out individual trends. As a result, the full range of the simulation ensemble was reported to reflect the uncertainty underlying the results.

L281: is this really 'the spatial pattern of trends' that is evaluated or is it a cell by cell comparison? Please elaborate and have in mind that although a correlation is it can still mean that the overall spatial pattern (i.e. approximate location of increasing and decreasing trends) might still be correct.

We assume that the reviewer means that "the overall spatial pattern of increasing and decreasing regions might still be correct even when the correlation value is low". During our investigation, we have conducted some visual inspections which confirmed that a low correlation value usually reflect the inconsistency in the spatial pattern between two specific set of trends (an example was provided in the Supplementary). This metric was also used extensively in the climate literature (Kumar et al., 2013;Kiktev et al., 2003;Kiktev et al., 2007) to assess the spatial consistency of trends introduced by different gridded products.

L 370-384: The authors mention 'a significant difference between trend characteristics from all model grid cells compared to those obtained from the observation locations' and the conclude that " that trends exhibited from observation locations are not a representative sample of trends obtained from all simulation grid cells" (L379-380) And then call "to improve data accessibility and expand streamflow observational networks". However, if there are such "significant difference even in data rich regions, how can one justify expanding the network based on the previous finding? Instead to me this reasoning would rather require the need to improve our models instead (notwithstanding the fact that I agree with the data needs mentioned by the authors.) We thank the reviewer for noting this out. We will carefully revise our discussion to incorporate this suggestion. Potential changes are (i) to elaborate more on model performance in data-rich regions, and (ii) highlight the need for improved capacity of GHMs in reproducing trends at the Conclusion.

We have revised the concluding remark of this paragraph into "it is therefore crucial to improve **not only models' capacity**, but also data accessibility and expand streamflow observational networks ..."

L 460: Maye the authors can elaborate a little more what an 'flexible adaptation strategy' entails in terms of flood mitigation. Any suggestion on how this can be achieved under tight budgets. Can we as scientists not provide any guidance than just saying 'stay flexible' to those who have to take decisions know?

We will consider extending our discussion to include feasible strategies and guidance to address high uncertainty in projections of changes in flood hazards.

We extended this statement to cover parts of the reviewer's comment.

L531 & 534: Along the lines of improved GHM: It is not only important that the spatial patterns are being reproduced correctly but also that the timing of the high-flows/floods are being modeled correctly. I.e. 'the flood seasonality patterns can be used as ' an additional metric to test large-scale hydro-logical models for their ability to reproduce the spatial and temporal flood characteristics.' (Hall and Bloschl, 2018, HESS). ' As this would give more confidence that the models actually get the flood generation processes correctly.

We thank the reviewer for this constructive comment. We agree that the timing of flood is a useful metric. This statistic should also be considered in the assessment of model capacity in terms of reproducing flood characteristics at the global and continental scale. We will extend our conclusion and include some corner-stone references (Hall and Blöschl, 2018;Blöschl et al., 2017;Dettinger and Diaz, 2000) to incorporate your suggestion.

We have revised our conclusion to incorporate the reviewer's comment.

**L 538: What does 'constraining ' entail? Please briefly elaborate. Would this prevent the model to adjust to changes in the flood generating processes, as one would expect to happen in some regions of the world. E.g. from snow-melt floods to rainfallgenerated floods?**

This term (i.e. "constraining") refers to the process of using observations to constrain multi-model projections and is commonly used in the climate literature (Padrón et al., 2017;Allen and Ingram, 2002). The purpose of this process is to prevent climate models projecting an unrealistic state of the future climate system (Flato et al., 2013). The constraints are usually the global average values of variables that model developers judge to be important (e.g. the global mean top of the atmosphere energy balance, cloud feedbacks). From our understanding, this process will not violate the fundamental physical processes of the hydrological cycle. We will clarify this terminology in the revision.

**Clarification added.**

L 550-559: I agree with this call, as this is very important. However, one needs to keep in mind that in many countries maintaining monitoring networks and data curation is/is considered too expensive. Hence it needs to be made clear to decision makers that such data is of importance. However, I know of cases where countries/agencies have been or are currently considering discontinuing their data networks, as they don't see the benefit or don't see their data being used (partly lack of proper citation of the (often freely available) original data source). This implication needs to be kept in mind when large datasets of observational data are being compiled and subsequently only credit is given to the compiled data. . . This hides to the funding/responsible agencies the usage of their data (i.e. the original data source) and might lead to the misconception that their data is not being needed/downloaded and hence the data network can be discontinued and to allocate funds to more (perceived) useful sectors. . .

We thank the reviewer for the comment. We agree this is very important to make national data authorities aware of the importance of their works. We will specifically emphasize the role of data "end-user" in making streamflow data more FAIR by properly acknowledging the efforts and merits that data providers deserve.

This paragraph (and the acknowledgement) has been revised.

**Fig S5: Suggest using same y-axis scale for all panels on the left/right to be able to compare the regions better with each another.**

We will revise the figure in our revision to ensure a consistent scale on the y-axis is used.

The figure has been revised.

- Abstract. To improve the understanding of trends in extreme flows related to flood events at the global scale, historical and future changes of annual maximum of 7-day streamflow are investigated, using a comprehensive streamflow archive and
- 27 Future enanges of annual maximum of ready streamnow are investigated, using a comprehensive streamnow are investigated.

[revised manuscript text omitted]

---

## Referee Report (RR1)

Dear authors,

I congratulate you to a very extensive and well-written manuscript. I am happy to see that all comments were addressed satisfactorily. Particularly the improved introduction, the overview of the GHMs used, and the csv-file in the supplement are very positively received.

I do not have any further comments and find the current version of the manuscript publishable.